# Ligand recognition mechanism of the human relaxin family peptide receptor 4 (RXFP4)

Yan Chen[1,14], Qingtong Zhou[1,14], Jiang Wang[2,3,4,14], Youwei Xu [5], Yun Wang[6], Jiahui Yan[5,7,8], Yibing Wang[2], Qi Zhu[6], Fenghui Zhao[5,7], Chenghao Li[2,4], Chuan-Wei Chen[9], Xiaoqing Cai[5,7], Ross A .D. Bathgate [10,11], Chun Shen[6], H. Eric Xu [5,8,12], Dehua Yang [5,7,8,9] ✉, Hong Liu [2,4,8,12] ✉ & Ming-Wei Wang [1,5,7,8,9,13] ✉

Members of the insulin superfamily regulate pleiotropic biological processes through two types of target-specific but structurally conserved peptides, insulin/insulin-like growth factors and relaxin/insulin-like peptides. The latter bind to the human relaxin family peptide receptors (RXFPs). Here, we report three cryo-electron microscopy structures of RXFP4–G$_i$ protein complexes in the presence of the endogenous ligand insulin-like peptide 5 (INSL5) or one of the two small molecule agonists, compound 4 and DC591053. The B chain of INSL5 adopts a single α-helix that penetrates into the orthosteric pocket, while the A chain sits above the orthosteric pocket, revealing a peptide-binding mode previously unknown. Together with mutagenesis and functional analyses, the key determinants responsible for the peptidomimetic agonism and subtype selectivity were identified. Our findings not only provide insights into ligand recognition and subtype selectivity among class A G protein-coupled receptors, but also expand the knowledge of signaling mechanisms in the insulin superfamily.

The human relaxin family peptide (RXFP) receptors (RXFP1, RXFP2, RXFP3 and RXFP4) play physiological roles through peptide hormones relaxin, insulin-like peptide 3 (INSL3), relaxin-3, and insulin-like peptide 5 (INSL5), respectively[1]. These peptides exert pleiotropic actions covering reproduction, cardiovascular adaptation, stress responses, metabolic control, colon motility, and behavioral processes[1], thereby showing therapeutic potential for a variety of disorders. Different from RXFP1 and RXFP2 that share a large extracellular domain containing 10

leucine-rich repeats (LRR) and a unique low-density lipoprotein class A module (LDL-A)[2–5], RXFP3, and RXFP4 have distinct binding properties with relatively short N-terminal tails rather than LRR. They possess 43% sequence identity in the overall structure and inhibit cAMP production via pertussis toxin-sensitive Gα$_{i/o}$ proteins[6].

RXFP4, also known as GPCR142 or GPR100, is primarily distributed in peripheral tissues with the highest expression in the colorectum[6,7]. Its endogenous ligand INSL5, secreted by the colonic

[1]Department of Pharmacology, School of Basic Medical Sciences, Fudan University, Shanghai 200032, China. [2]State Key Laboratory of Drug Research, Shanghai Institute of Materia Medica, Chinese Academy of Sciences, Shanghai 201203, China. [3]Lingang Laboratory, Shanghai 200031, China. [4]School of Pharmaceutical Science and Technology, Hangzhou Institute for Advanced Study, UCAS, Hangzhou 310024, China. [5]The CAS Key Laboratory of Receptor Research, Shanghai Institute of Materia Medica, Chinese Academy of Sciences, Shanghai 201203, China. [6]Genova Biotech (Changzhou) Co., Ltd, Changzhou 213125, China. [7]The National Center for Drug Screening, Shanghai Institute of Materia Medica, Chinese Academy of Sciences, Shanghai 201203, China. [8]University of Chinese Academy of Sciences, Beijing 100049, China. [9]Research Center for Deepsea Bioresources, Sanya, Hainan 572025, China. [10]The Florey Institute of Neuroscience and Mental Health, University of Melbourne, Parkville, Victoria 3052, Australia. [11]Department of Biochemistry and Molecular Biology, University of Melbourne, Parkville, Victoria 3052, Australia. [12]School of Life Science and Technology, ShanghaiTech University, Shanghai 201210, China. [13]Department of Chemistry, School of Science, The University of Tokyo, Tokyo 113-0033, Japan. [14]These authors contributed equally: Yan Chen, Qingtong Zhou, Jiang Wang. ✉e-mail: dhyang@simm.ac.cn; hliu@simm.ac.cn; mwwang@simm.ac.cn

L-cell, was originally identified as an incretin albeit with some controversies[6–9]. Their expression pattern together with impaired glucose and fat control shown in INSL5 or RXFP4 deficient mice indicate their involvement in energy metabolism[6,7,10,11]. INSL5 has also been described as an orexigenic hormone[12] and RXFP4 was implicated in colon motility[13,14], colorectal cancer, and nasopharyngeal carcinoma[15,16].

Despite these advances, difficulties in obtaining sufficient quantities of native INSL5 hampered our efforts in further exploring the biology of the peptide and its cognate receptor. Since relaxin-3 also binds to and activates RXFP4 in vitro[6], it has been used as a surrogate ligand to study potential actions of INSL5 due to their shared tertiary structure closely related to insulin including two chains and three disulfide bonds[17]. In addition to peptidic analogs, small molecule modulators have been reported in recent years. Compound 4, an amidino hydrazone-based scaffold identified by Novartis, is an RXFP3/RXFP4 dual agonist[18]. In vivo, the overlapping expression pattern between RXFP4 and RXFP3 as well as their distinct physiological properties[19,20] call for subtype-specific agonists which will likely be valuable to different clinical applications. However, selective RXFP4 agonists discovered via high-throughput screening campaigns and follow-up structural modifications displayed deficiencies in solubility, potency, and toxicity[21,22]. This promoted us to develop a small molecule agonist (DC591053) with better affinity and selectivity for RXFP4.

In this work, we report three cryogenic electron microscopy (cryo-EM) structures of the human RXFP4–G$_i$ complexes bound to INSL5, compound 4, and DC591053 with global resolutions of 3.19 Å, 3.03 Å, and 2.75 Å, respectively. Together with mutagenesis and functional analyses, we describe a peptide-binding mechanism previously unseen in other class A G protein-coupled receptors (GPCRs) and provide useful information for structure-based design of RXFP4 agonists either as research probes or as drug candidates.

## Results

### Characterization of recombinant INSL5
The purity of recombinant INSL5 was over 90% by reverse phase high-performance liquid chromatography (RP-HPLC) and the molecular weight was determined to be 5061.2 Da by mass spectrometry (MS), equivalent to that of native INSL5 peptide (5062.9 Da; N-terminal Q of A chain not converted to pE) (Supplementary Fig. 1). As depicted in Supplementary Fig. 1, chymotrypsin cleavage resulted in 8 major peaks (labeled as ①-⑧) on RP-HPLC. The measured molecular masses of the individual peaks were identical to the theoretical values of the expected chymotrypsin-generated peptides, which allowed for 100% sequence coverage. The recombinant INSL5 peptide was subsequently verified for its bioactivity in Chinese hamster ovary (CHO-K1) cells stably transfected with RXFP4. As shown in Supplementary Fig. 1, it bound to RXFP4 with high affinity and was able to inhibit forskolin-induced cAMP responses ($p$EC$_{50}$ = 8.80 ± 0.11, $n$ = 3; $p$Ki = 8.19 ± 0.06, $n$ = 3, as measured in stably-transfected CHO-K1 cells) compared to the native INSL5 standard.

### Characterization of DC591053
We screened our in-house tetrahydroisoquinoline library aimed at discovering RXFP4 agonists using cAMP accumulation assay. Of the six compounds displaying RXFP4 agonist activities (data not shown), DC591053 ((S)-(7-ethoxy-6-methoxy-1-(2-(5-methoxy-1H-indol-3-yl)ethyl)−3, 4-dihydroisoquinolin-2(1H)-yl)(morpholino)methanone) exhibited the best agonism. It was synthesized from the commercially available compound 4-hydroxy-3-methoxybenzaldehyde, followed by alkylation reaction, reduction, Wittig reaction, cyclization, asymmetric reduction reaction, and condensation reaction (Supplementary Fig. 2a–d). DC591053 demonstrated full agonism at RXFP4 both in competitive europium (Eu)-labeled R3/I5 binding and cAMP accumulation assays ($p$EC$_{50}$ = 7.24 ± 0.12, $n$ = 3; $p$Ki = 6.95 ± 0.14, $n$ = 3, as measured in stably-transfected CHO-K1 cells). Importantly, DC591053 neither reacted with related RXFP3 nor with parental CHO-K1 cells (Supplementary Fig. 2e–g).

### Overall structures
To prepare a high-quality human RXFP4–G$_i$ complex, we added a haemagglutinin (HA) signal peptide to enhance receptor expression, followed by a 10× histidine tag as well as cytochrome b562RIL (BRIL) insertion at the N terminus, and applied the NanoBiT tethering strategy (Supplementary Fig. 3a)[23–25]. The activity of the modified RXFP4 construct was confirmed by cAMP accumulation assay showing a response similar to that of the wild-type (WT). These complexes were then purified, resolved as monodispersed peaks on size-exclusion chromatography (SEC), and verified by SDS gel to ascertain all the expected components (Supplementary Fig. 3b–d). After sample preparation, cryo-EM data were collected, analyzed and 3-dimensional (3D) consensus density maps reconstructed (Supplementary Fig. 4) resulting in an overall resolution of 3.19 Å, 3.03 Å, and 2.75 Å for the INSL5–RXFP4–G$_i$, compound 4–RXFP4–G$_i$ and DC591053–RXFP4–G$_i$ complexes, respectively (Fig. 1, Supplementary Table 1). These maps allowed us to build near-atomic level models for most regions of the complexes except for the flexible α-helical domain (AHD) of G$_i$, the N terminus (M1 to K34) and the intracellular loop 1 (ICL1) between N66 to P72 of RXFP4 (Fig. 1, Supplementary Fig. 5). Because of the relatively high resolution of the three structures, the RXFP4-bound INSL5, compound 4 and DC591053 were well-defined in the EM density maps.

These structures share a similar conformation with root mean squared deviation (RMSD) of <0.5 Å, including a hallmark outward movement of the intracellular half of transmembrane helix (TM) 6 relative to the X-ray structures of inactive β$_2$-adrenergic receptor or cholecystokinin A receptor (CCK$_A$R)[26–28] (Supplementary Fig. 6) and a β-hairpin occurred in the second extracellular loop (ECL2) that is similar to the peptide-bound class A GPCR structures such as CCK$_A$R[27], cholecystokinin B receptor (CCK$_B$R)[27], type 1 bradykinin receptor (B1R)[29], type 2 bradykinin receptor (B2R)[29] and C-C chemokine receptor type 1 (CCR1)[30]. One significant difference is that INSL5 displayed a previously unknown binding mode to the cognate receptor (Supplementary Fig. 7): its C-terminal α-helix of the B chain penetrated into the transmembrane domain (TMD) core, such that the two terminus residues R23$^B$ (B indicates that the residue belongs to the B chain of INSL5) and W24$^B$ fully occupied the orthosteric pocket, while the A chain strengthened the binding by restraining the movement of the B chain through two inter-chain disulfide bonds (C8$^A$–C7$^B$ and C21$^A$–C19$^B$) (Fig. 2a). Both compound 4 and DC591053 displayed a peptidomimetic feature by structurally and spatially mimicking the C-terminal tryptophan (W24$^B$) as a common chemotype; with ligand-specific recognition by TM5, TM7 and ECL2 to confer distinct subtype selectivity of RXFP4 over RXFP3 (Fig. 3a, f, Supplementary Table 2).

### Peptide recognition
INSL5 anchored in the RXFP4 orthosteric binding pocket bordered by TMs 2-7 and ECLs 1-3, with its B chain inserting into the TMD bundle and contributing a majority of the receptor interaction sites, while the A chain docked above the orthosteric pocket and interacted with ECL2, ECL3, and solvent (Fig. 2a–e). Consistently, the interface area between RXFP4 and the B chain (1444 Å$^2$) is significantly larger than that of the A chain (351 Å$^2$).

The B chain of INSL5 exhibited a single amphipathic α-helix conformation[31] from E10$^B$ to W24$^B$, with the C terminus W24$^B$ being the deepest residue in the receptor core. The N-terminal residues (R5$^B$ to L9$^B$) adopted a loop that is clasped by a short N-terminal α-helix of the A chain through one disulfide bond (C8$^A$–C7$^B$). W24$^B$ contributed massive polar and nonpolar interactions to stabilize the peptide binding via both side chain indole and the carboxylic acid group. The former made a hydrogen bond with T121$^{3.32}$ (superscripts denote Ballesteros–Weinstein numbering[32]), as well as cation-π

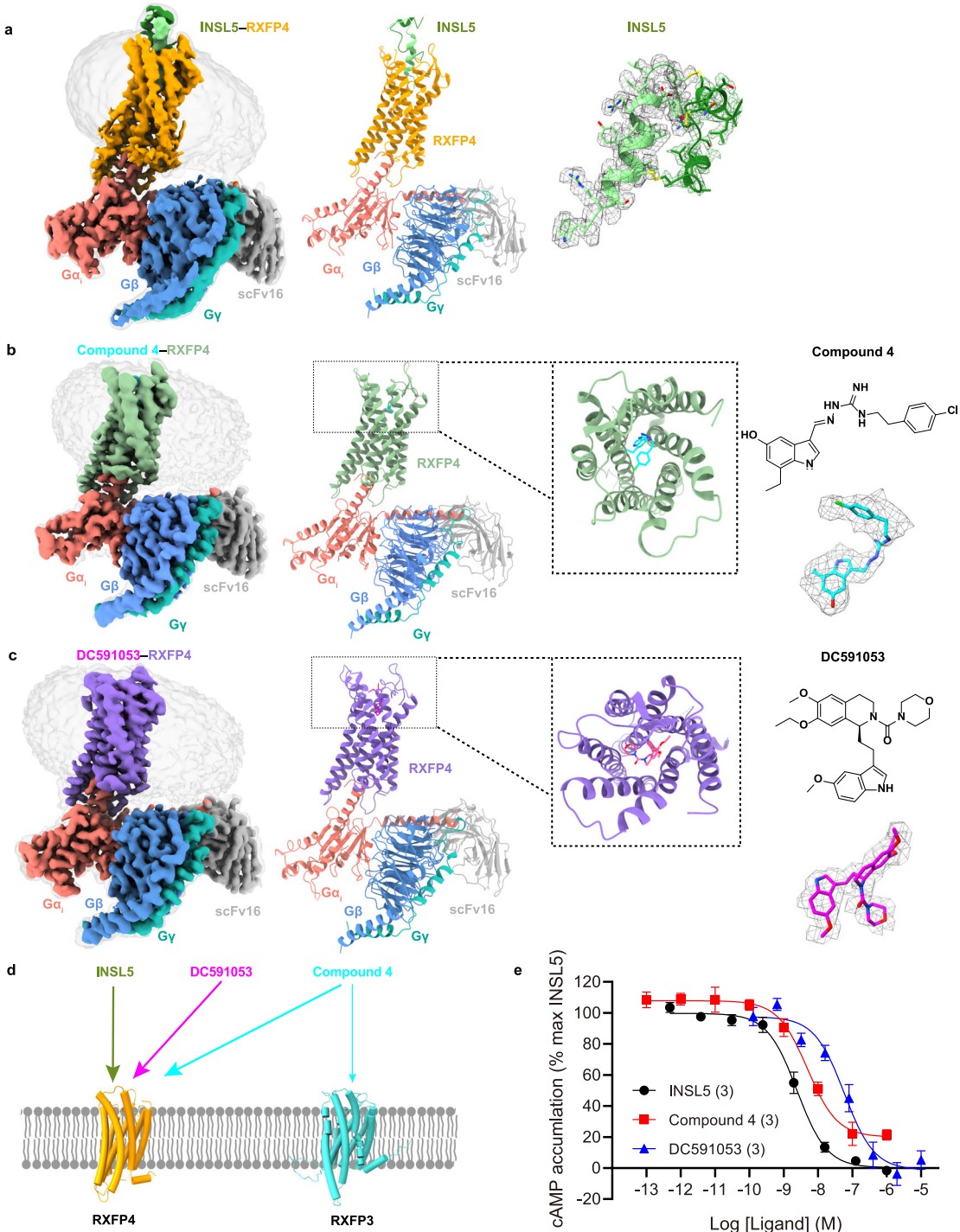

**Fig. 1 | Cryo-EM structures of the RXFP4-G_i complexes. a–c** Cryo-EM density maps (left panel) and cartoon representation (middle) of the INSL5−RXFP4−G_i−scFv16 complex (**a**), compound 4−RXFP4−G_i−scFv16 complex (**b**) and DC591053−RXFP4−G_i−scFv16 complex (**c**). Atomic models and EM densities of the three ligands are shown as sticks and surfaces, respectively. The A chain of INSL5 is shown in forest green and the B chain in light green, compound 4 in cyan and DC591053 in magenta. The corresponding RXFP4 is shown in orange, dark sea green and medium purple, respectively. Gα_i in salmon, Gβ in cornflower blue, Gγ in light sea green, and scFv16 in dark gray. **d** Activities of INSL5 and DC591053 in RXFP4 and cross-reactivity of compound 4 in RXFP3 and RXFP4. The thickness of lines indicates the strength of affinity. **e** INSL5, compound 4 and DC591053 induced cAMP signaling in cells expressing wild-type RXFP4. Data shown are means ± S.E.M. of three independent experiments (*n* = 3) performed in quadruplicate. Source data are provided as a Source Data file.

stacking with R208^5.42 and π-π stacking with W97^2.60, F291^7.35, and H299^7.43, while the latter pointed to TM5 with the formation of one hydrogen bond (via Q205^5.39) and one salt bridge (via R208^5.42) (Fig. 2b). These observations support the importance of a free carboxyl group in the B chain C terminus for high-affinity RXFP4 binding and signaling activity[33], consistent with our mutagenesis

studies showing that INSL5-induced cAMP responses were completely abolished in mutants T121^3.32A and R208^5.42A, profoundly reduced in mutant H299^7.43A (E_max value by 70%) or markedly diminished in mutants W97^2.60A and Q205^5.39A by 20.4-fold and 5.3-fold, respectively (Fig. 2f–g, Supplementary Table 4). In addition, alanine replacement of W24^B and amidation of the B chain C

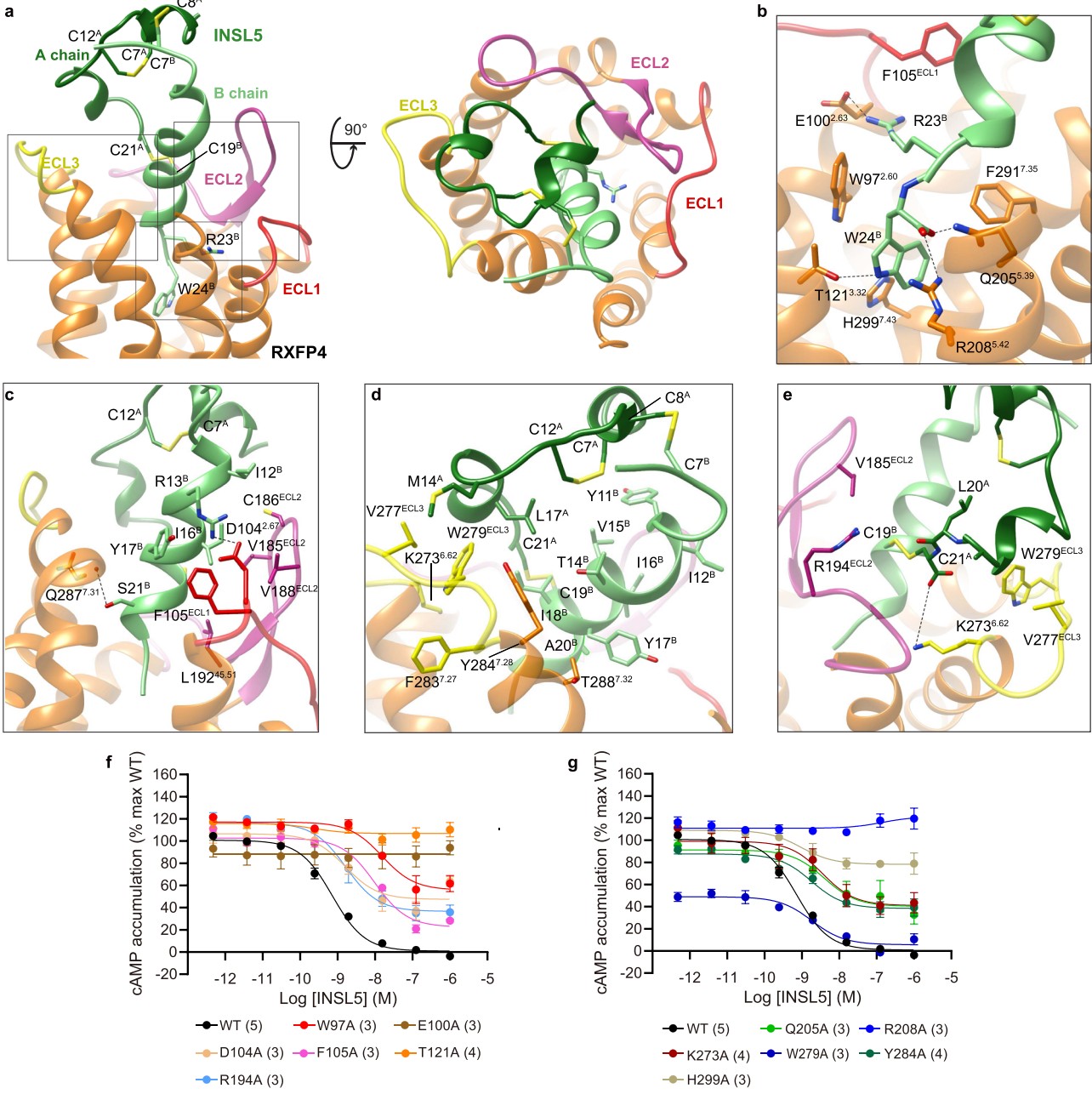

**Fig. 2 | Molecular recognition of INSL5 by RXFP4. a** Binding mode of INSL5 (green) with RXFP4 (orange), showing that the B chain of INSL5 (light green) penetrates into the transmembrane domain (TMD) bundle using the C-terminal α-helix, whereas the A chain of INSL5 (forest green) covers the orthosteric pocket mainly interacting with the extracellular loops 2 (ECL2, violet red) and 3 (ECL3, yellow). **b** Interactions of R23[B] and W24[B] with RXFP4 in the orthosteric pocket. **c**–**e** Close-up views of the interactions between the B chain of INSL5 and the residues in ECLs of RXFP4. **f**, **g** Effects of receptor mutations on INSL5-induced cAMP accumulation. Data are shown as means ± S.E.M. of at least three independent experiments. The numbers of independent experiments are shown in parentheses. Supplementary Table 4 provides detailed statistical evaluation such as *P* values and numbers of the independent experiment (*n*). Source data are provided as a Source Data file.

terminus significantly reduced INSL5-elicited agonistic activity as previously described[34]. Another important residue is R23[B] whose side chain oriented towards TM2 and formed one salt bridge with E100[2.63]. Mutation of E100[2.63] to alanine (Fig. 2f, Supplementary Table 4) or arginine[33] both deprived the ability of INSL5 to activate RXFP4, in line with the reduced potencies reported for R23[B]A[34] or R23[B]E[35]. Interestingly, diverse peptide-receptor contacts were observed for the residue at 2.63 depending on physicochemical properties including positively charged [e.g., R102[2.63] in growth hormone secretagogue receptor (GHSR) and R84[2.63] in formyl peptide receptor 1 (FPR1)] and negatively charged amino acids [e.g.,

D93[2.63] in C-X-C chemokine receptor type 4 (CXCR4)]. Besides the polar contacts, R13[B] and S21[B] made one salt bridge and one hydrogen bond with the side chain of D104[2.67] and backbone oxygen of Q287[7.31], respectively, while Y17[B] was stabilized by the π-π stacking from F105[ECL1] (Fig. 2c, d). The hydrophobic residues in the B chain further strengthened INSL5 binding by hydrophobic contacts with both RXFP4 residues (F105[ECL1], V185[ECL2], C186[ECL2], V188[ECL2], L192[45.51], K273[6.62], W279[ECL3] and Y284[7.28]) and the A chain residues (L3[A] and L17[A]) via Y11[B], V15[B], I16[B], I18[B], C19[B] and A20[B] (Fig. 2c, d). Disruption of these hydrophobic contacts through mutants F105[ECL1]A, K273[6.62]A, W279[ECL3]A and Y284[7.28]A moderately decreased both potency and

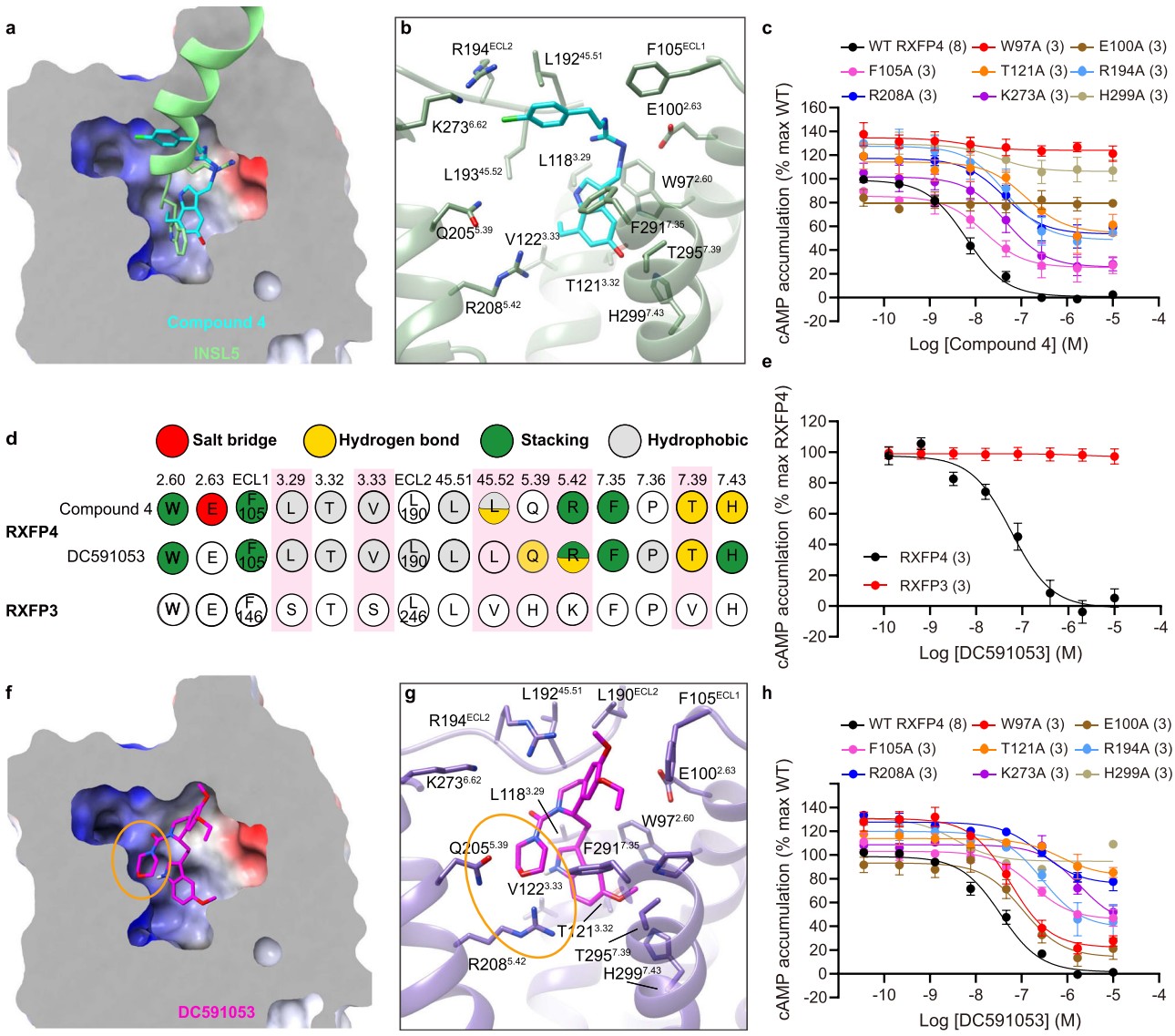

**Fig. 3 | Peptidomimetic agonism and subtype selectivity demonstrated by compound 4 and DC591053. a** The cross-section view of the compound 4 binding pocket in RXFP4. **b** Detailed interactions of compound 4 (cyan) with residues in RXFP4 (dark sea green). **c** Effects of receptor mutations on compound 4-induced cAMP accumulation. Data are shown as means ± S.E.M. of at least three independent experiments. **d** Schematic diagram of interactions between ligands and receptor. Amino acid residues of RXFP4 are colored red for salt bridge, yellow for hydrogen bond, green for π-π stacking and gray for hydrophobic interactions. Different residues around the binding pocket in RXFP3 and RXFP4 are highlighted in pink. **e** Subtype selectivity of DC591053 at RXFP4 without observable cross-reactivity in RXFP3. Data are shown as means ± S.E.M. of three independent experiments (*n* = 3). **f** The cross-section view of the DC591053 binding pocket in RXFP4, with the RXFP4-specific edge in the ligand-binding pocket highlighted in an orange circle. **g** Detailed interactions of DC591053 (magenta) with residues in RXFP4 (medium purple). **h** Effects of receptor mutations on DC591053-induced cAMP accumulation. Data are shown as means ± S.E.M. of at least three independent experiments. The numbers of independent experiments are shown in the parentheses. Supplementary Table 4 provides detailed statistical evaluation such as *P* values and numbers of the independent experiment (*n*). Source data are provided as a Source Data file.

$E_{max}$ of INSL5 (Fig. 2f, g, Supplementary Table 4), supported by weak or moderate decreases in binding affinity and potency when the B chain residues I12[B], V15[B], I16[B] and I18[B] were mutated to alanine[34]. Consistently, molecular dynamics (MD) simulations found that the C-terminal α-helix of the B chain could stably maintain its insertion into the orthosteric pocket through its tip residues, evidenced by the interface area and representative minimum distances (R13[B]–D104[2.67], R23[B]–E100[2.63] and W24[B]–Q205[5.39]/R208[5.42]) (Supplementary Fig. 8a–i). Notably, the internal water molecules were found to fill the orthosteric pocket with the formation of multiple contacts with surrounding polar residues in both RXFP4 and the C terminus of INSL5 B chain during MD simulations (Supplementary Fig. 8j, k) as seen in other GPCRs[36–38].

Different from the binding mode of B chain that was largely buried by the TMD bundle, A chain solely interacted with several residues in ECL2 and ECL3 forming one salt bridge (via the side chain of K273[6.62]), one hydrogen bond (via the side chain of R194[ECL2]) and multiple hydrophobic contacts (via V185[ECL2], V277[ECL3], and W279[ECL3]) (Fig. 2e). As expected, alanine substitutions at K273[6.62], R194[ECL2] and W279[ECL3] modestly reduced INSL5 potency by 4.7-fold, 2.2-fold and 2.5-fold, respectively (Fig. 2f, g, Supplementary Table 4). Instead of direct interaction with RXFP4, A chain is likely to stabilize peptide binding by restraining the dynamics of INSL5 through three disulfide bonds and a hydrophobic patch (L3[A], L6[A], L17[A], L20[A], Y11[B], V15[B], and I18[B]), thereby maintaining the correct conformation of INSL5 for RXFP4 recognition and reducing the entropy cost during peptide binding. Functional and

MD simulation studies are in agreement with this observation as deletion of the A chain completely abolished receptor binding and signaling activities of INSL5[39] (Supplementary Fig. 9), suggesting that B chain alone is not sufficient to sustain the α-helix conformation.

## Receptor selectivity

Strong electron densities were observed for compound 4 from the orthosteric site of RXFP4 to ECL2, revealing a C-shaped conformation of compound 4, with the indole ring inserting deeply into the orthosteric binding pocket and its chlorobenzene moiety extending to the extracellular side (Fig. 3a). By displaying a conformation similar to the C-terminal residue $W24^B$ of INSL5, the indole ring of compound 4 showed strong interactions with RXFP4 residues, forming two hydrogen bonds (via $T295^{7.39}$ and $H299^{7.43}$), stacking contacts (via $W97^{2.60}$, $R208^{5.42}$, and $F291^{7.35}$) and hydrophobic contacts (via $L118^{3.29}$, $T121^{3.32}$, and $V122^{3.33}$). The central guanidine moiety was positively charged to mimic $R23^B$ of INSL5 and made one salt bridge with the negatively charged side chain of $E100^{2.63}$ as well as cation-π stacking interactions with $F105^{ECL1}$. The chlorobenzene group covered the orthosteric site and was close to ECL2 with the formation of multiple hydrogen bonds (via the backbone oxygen atom of $L193^{45.52}$ and $R194^{ECL2}$) and hydrophobic contacts (via $L192^{45.51}$ and $L193^{45.52}$) (Fig. 3b). Mutagenesis and structure-activity relationship (SAR) studies support these observations: mutants $W97^{2.60}A$, $E100^{2.63}A$, and $H299^{7.43}A$ abolished cAMP responses, while $T121^{3.32}A$ and $R208^{5.42}A$ significantly impaired the potency of compound 4 by 20.1-fold and 6.6-fold, respectively (Fig. 3c, Supplementary Table 4); substitution of hydroxy by methoxyl at the indole 5 position or replacement of ethyl by a smaller methyl at the indole 7 position eliminated the hydrogen bonds with TM7 residues and weakened hydrophobic contacts with TM3 residues, respectively, thereby reducing the agonist potencies as reported previously[18].

Since the sequence identity of the ligand-binding pocket between RXFP3 and RXFP4 is 86.36%, the development of receptor subtype-selective ligands is very challenging. Only six pocket residues are diversified: $S159^{3.29}$, $S163^{3.33}$, $V249^{45.52}$, $H268^{5.39}$, $K271^{5.42}$, and $V375^{7.39}$ for RXFP3, and $L118^{3.29}$, $V122^{3.33}$, $L193^{45.52}$, $Q205^{5.39}$, $R208^{5.42}$, and $T295^{7.39}$ for RXFP4. Compound 4 formed one hydrogen bond with the side chain of $T295^{7.39}$ which is unlikely to occur in the equivalent position of RXFP3 ($V375^{7.39}$). However, two distinct amino acids in TM5 ($Q205^{5.39}$, $R208^{5.42}$ for RXFP4 and $H268^{5.39}$, $K271^{5.42}$ for RXFP3[40]) were not contacted, which may limit the subtype selectivity. To overcome this hurdle, DC591053 was developed to demonstrate a full agonism at RXFP4 ($pEC_{50} = 7.24 \pm 0.12$) without observable cross-reactivity with RXFP3 (Fig. 3e).

As shown in Fig. 3f, the indole ring of DC591053 occupied the orthosteric pocket in a similar manner as $W24^B$ of INSL5 and compound 4. It also stabilized the RXFP4−$G_i$ complex by stacking interactions with $W97^{2.60}$, $R208^{5.42}$, $F291^{7.35}$, and $H299^{7.43}$ as well as hydrophobic contacts with $L118^{3.29}$, $T121^{3.32}$, and $V122^{3.33}$ (Fig. 3g). Mutants $W97^{2.60}A$ and $T121^{3.32}A$ suppressed the ability of RXFP4 to inhibit cAMP production upon DC591053 stimulation (by 1.6-fold and 20.9-fold, respectively), and $H299^{7.43}A$ seriously affected the $E_{max}$ value (reduced by 65%) (Fig. 3h, Supplementary Table 4). The methoxyl at the indole 5 position of DC591053 pointed towards TM7 with the formation of one hydrogen bond (via $T295^{7.39}$). Different from compound 4, the morpholine ring rendered DC591053 to form two moderate hydrogen bonds with $Q205^{5.39}$ and $R208^{5.42}$, i.e., an RXFP4-specific edge in the ligand-binding pocket, which may enhance the selectivity for RXFP4 (Fig. 3f, g). Consistently, $R208^{5.42}A$ decreased the potency of DC591053 by 7.6-fold (Fig. 3h, Supplementary Table 4). Another notable difference is the replacement of guanidine moiety in compound 4 and $R23^B$ of INSL5 by the urea group in DC591053, which is unlikely to make polar interaction with $E100^{2.63}$, in agreement with unchanged agonism of DC591053 at mutant $E100^{2.63}A$ whose signaling

is abolished for INSL5 and compound 4. To compensate for the contact gap caused by the above replacement, the tetrahydroisoquinoline moiety of DC591053 contributed multiple stacking interactions with $F105^{ECL1}$, $R194^{ECL2}$, and $F291^{7.35}$ and hydrophobic contacts with $L190^{ECL2}$, $L192^{45.51}$ and $P292^{7.36}$ (Fig. 3g), which are significantly stronger than that of compound 4. Removal of these contacts by mutants $F105^{ECL1}A$ and $R194^{ECL2}A$ reduced DC591053 potency by 4.9-fold and 8.1-fold, respectively (Fig. 3h, Supplementary Table 4). To further explore subtype selectivity, we performed amino acid switch studies in the equivalent positions between RXFP4 and RXFP3 around the ligand-binding pocket. Double mutant $L118^{3.29}S + V122^{3.33}S$ in RXFP4 selectively affected the potency of DC591053 by 20.9-fold without notable influence on that of compound 4. As a comparison, $S159^{3.29}L + S163^{3.33}V$ in RXFP3 reduced the potency of compound 4 by 26.9-fold. Similar phenomena were also observed in $Q205^{5.39}H$ and $R208^{5.42}K$ in RXFP4 (displayed more profound reduction for DC591053 than compound 4), while $H268^{5.39}Q$ and $K271^{5.42}R$ in RXFP3 exhibited dose-response features for compound 4 similar to the WT (Supplementary Fig. 10b–d, Supplementary Table 4). Notably, mutations at $S159^{3.29}$, $S163^{3.33}$, and $V375^{7.39}$ in RXFP3 and $L118^{3.29}$, $V122^{3.33}$, and $T295^{7.39}$ in RXFP4 caused differentiated influences on the potencies of INSL5 and relaxin-3 (Supplementary Fig. 10e–g, Supplementary Table 5). The results indicate that these sites may play important roles in subtype selectivity.

## $G_i$ coupling

$G_i$-coupling was almost identical among the three complex structures (Fig. 4a), where $G_i$ protein was anchored by the α5 helix of $G_i$ subunit, thereby fitting to the cytoplasmic cavity formed by TMs 2, 3 and 5–7 as well as ICLs 2 and 3, a phenomenon widely observed in other $G_i$-coupled structures such as GHSR[41], formyl peptide receptor 2 (FPR2)[42] and CCR1[30] (Fig. 4a, b). The hydrophobic patch at the C terminus of $G_i$, including $I345^{G.H5.16}$ (superscripts refer to the common Gα numbering system[43]), $L349^{G.H5.20}$, $C352^{G.H5.23}$, $L354^{G.H5.25}$, and $F355^{G.5.26}$, interacted with a series of surrounding hydrophobic residues in TMs 3, 5, and 6 by contributing massive hydrophobic contacts (via $V142^{3.53}$, $V143^{3.54}$, $Y224^{5.58}$, $L227^{5.61}$, $F230^{5.64}$, $L231^{5.65}$, $V243^{6.32}$, $V244^{6.33}$, $V248^{6.37}$, and $L251^{6.40}$), three hydrogen bonds ($R139^{3.50}–C352^{G.H5.23}$, $V142^{3.53}–N348^{G.H5.19}$, and $S247^{6.36}–L354^{G.H5.25}$) and one salt bridge ($D240^{6.29}–K346^{G.H5.17}$) (Fig. 4c). Unlike the short α-helix conformation that observed in FPR2, CCR1 and somatostatin receptor 2 (SSTR2), ICL2 of RXFP4 adopted a loop conformation and made one hydrogen bond ($H152^{ICL2}–N348^{G.H5.19}$) and multiple hydrophobic contacts via $A147^{ICL2}$ and $P149^{ICL2}$ with $G_i$ (Fig. 4d). Consistent with the crucial role of ICL3 in signaling pathways of various GPCRs[44–46], three adjacent positively charged residues ($R234^{ICL3}$, $R236^{ICL3}$, and $R237^{ICL3}$) and $Q235^{ICL3}$ established a polar network through multiple salt bridges (via $E309^{G.H4.26}$, $E319^{G.h4s6.12}$ and $D342^{G.H5.13}$) and several hydrogen bonds (via $D338^{G.H5.9}$, and $T341^{G.H5.12}$) (Fig. 4e). Notably, one salt bridge between helix 8 and α5 helix of $G_i$ ($E315^{8.49}–K350^{G.H5.21}$) was found only in the cryo-EM structure of compound 4−RXFP4−$G_i$ complex (Fig. 4a).

## Class-wide comparison

Endogenous peptides mainly bind to class A and B1 GPCRs[47,48]. Unlike its class B1 counterparts that have large extracellular domains, class A GPCRs usually adopt extended loop conformations during their insertion into the orthosteric pocket by the peptide N terminus [*e.g.*, DAMGO[49], C-C chemokine ligand 15 (CCL15)[30], C-X-C motif chemokine ligand 8 (CXCL8)[50], $Aβ_{42}$[51], *N*-formyl humanin[51] and ghrelin[41]], the peptide C terminus [*e.g.*, angiotensin II[52,53], bradykinin[29], cholecystokinin-8 (CCK-8)[54], Des-Arg[10]-kallidin[29], gastrin-17[27], JMV449[55], neuromedin U[56], and neuromedin S[56]] or the peptide middle region [*e.g.*, α-melanocyte-stimulating hormone (α-MSH)[57], arginine-vasopressin (AVP)[58] and somatostain-14[59]], thereby achieving a significantly larger peptide-receptor interface area (>1500 Å²) compared

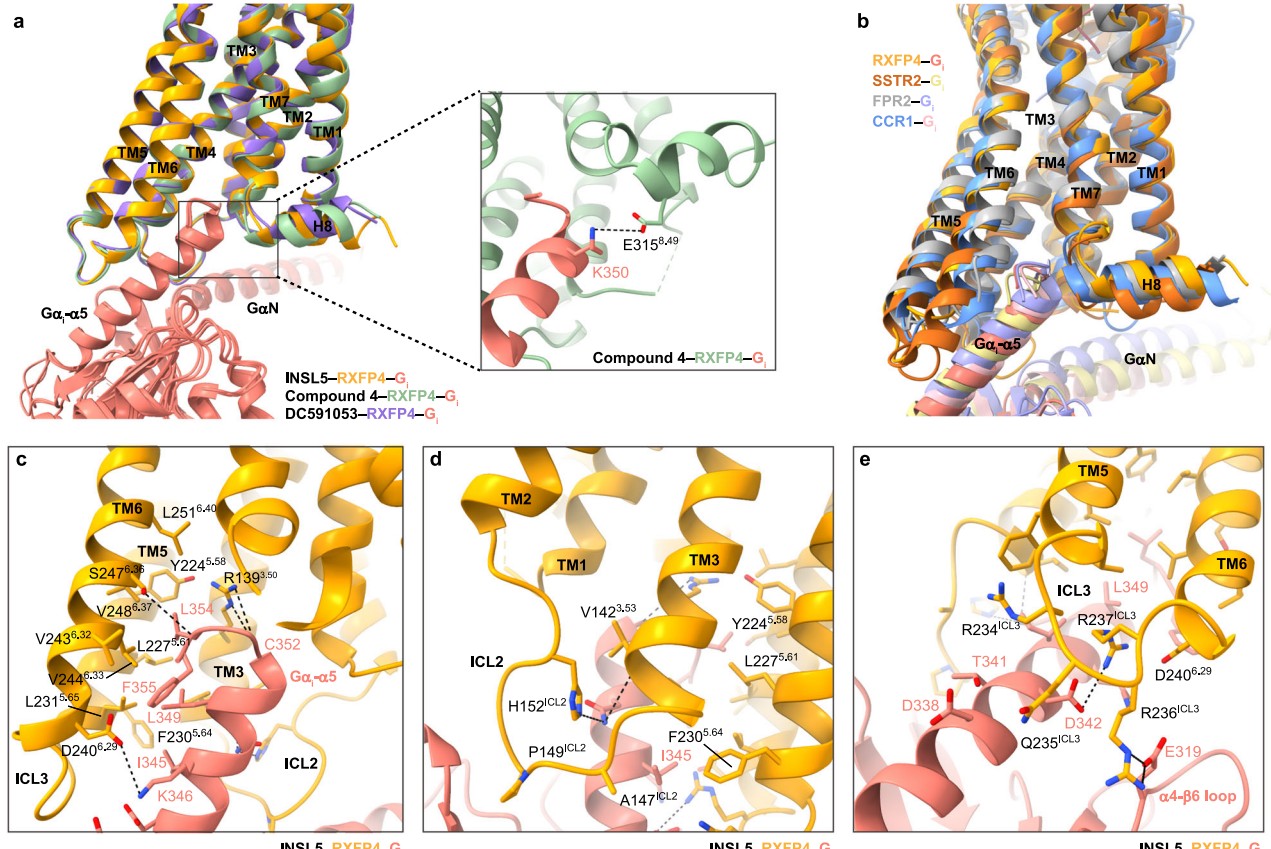

**Fig. 4 | G protein coupling of RXFP4. a** $G_i$-coupling is almost identical among INSL5-, compound 4- and DC591053-bound RXFP4 structures. The alignment is based on the receptor. One salt bridge between helix 8 (H8) and α5 helix of $G_i$ was only found in the compound 4-bound RXFP4-$G_i$ complex. **b** Comparison of G protein coupling among INSL5-bound RXFP4, FPR2 (PDB code: 7WVV), CCR1 (PDB code: 7VL9) and SSTR2 (PDB code: 7T10). The receptors and G proteins were colored as labeled. **c** Interaction between RXFP4 (orange) and α5 helix of $G_i$ (salmon) in the cavity of the cytoplasmic region. **d** Interactions between intracellular loop 2 (ICL2) and $G_i$. **e** Interactions between ICL3 and $G_i$. Polar interactions are shown as black dashed lines.

to that displayed by interaction with small molecules (<1000 Å²) (Fig. 5, Supplementary Fig. 7). Of note, galanin, located far away from the receptor core[60,61], adopted an α-helical structure that sat flat on the top of the orthosteric pocket with formation of massive contacts with ECLs 1-3 and moderate interface area (~1600 Å²). Different from the above peptide-binding modes, INSL5 penetrates into the orthosteric pocket via its B chain C terminus by adopting a single α-helix conformation, which is distinct from all reported peptide-bound class A GPCRs but closer to those seen with class B1 structures bound by peptides, such as glucagon-like peptide-1 (GLP-1), glucose-dependent insulinotropic polypeptide (GIP) and glucagon whose N termini insert deeply into the TMD core. This organization resulted in a profound interface area (1761 Å²) for INSL5 and direct signal initiation via engagement of α-helix terminus W24^B. Obviously, this α-helix conformation was maintained by the three disulfide bonds, supported by the conserved three helical segments of INSL5 observed in solution-state NMR studies[31].

### Mechanistic implication

Sharing the same structural scaffold (three α-helices constrained by one intra- and two inter-chain disulfide bonds) and the insulin signature (CC-3X-C-8X-C motif in the A chain), insulin, insulin-like growth factors (IGFs) 1 and 2, relaxins 1–3 and INSL3-6 constitute the human insulin superfamily (Fig. 6a), an ancient family of functionally diverse proteins[62,63]. While insulin and IGF-1 mainly bind to and activate cell surface tyrosine kinase receptors, i.e., canonical insulin receptor (IR)/ IGF-1 receptor (IGF-1R), and IGF-2 acts through the single-transmembrane glycoprotein IGF-2/mannose-6-phosphate receptor (IGF-2R/M6PR); the actions of relaxins 1–3, INSL3 and INSL5 are

mediated by respective GPCRs. The INSL5-bound RXFP4−$G_i$ complex structure, together with abundant information on insulin and IGFs in the literature[64–67], provides an excellent opportunity to investigate the structural basis of the functional versatility with no cross-reactivity among members of this important peptide superfamily.

The peptide-binding pocket of RXFP4 is significantly different from that of the insulin and IGF-1 receptors. By arranging the residues at the extracellular halves of TMs 2-7, RXFP4 provides a typical class A GPCR pocket that is deeply buried and occluded for the penetration of the C-terminal α-helix of INSL5 B chain (α1 in Fig. 6b). Meanwhile, the ECLs of RXFP4 interact with the C-terminal region of the second short α-helix of INSL5 A chain (α3 in Fig. 6b). Such a binding mode suggests that the sequence and length at the C-terminal ends of A and B chains are likely to play a key role in receptor activation and subtype selectivity. Consistently, the C-terminal truncation at the B chain of relaxin-2 greatly reduced agonist potency by 100-fold compared to the native peptide[68]. Such a truncation transformed relaxin-3 to an antagonist for RXFP3 and RXFP4[69]. Because of the presence of additional residues at the C termini of both chains, insulin and IGFs produced massive sterically clashes with RXFP4 upon structure superimposition (Supplementary Fig. 11a, b), implying that they are unable to bind and activate RXFPs. As a comparison, the binding pockets of IR and IGF-1R are planar and largely solvent-exposed, where distinct segments of the conserved structural feature were used by insulin or IGF-1 for receptor recognition (Fig. 6c, d)[70]. Specifically, both peptides utilized the hydrophobic residues at the two short α-helices (α2 and α3) as hydrophobic core to interact with the hydrophobic residues in IR and IGF-1R, whereas the extended C-terminal tail of insulin's B chain sealed

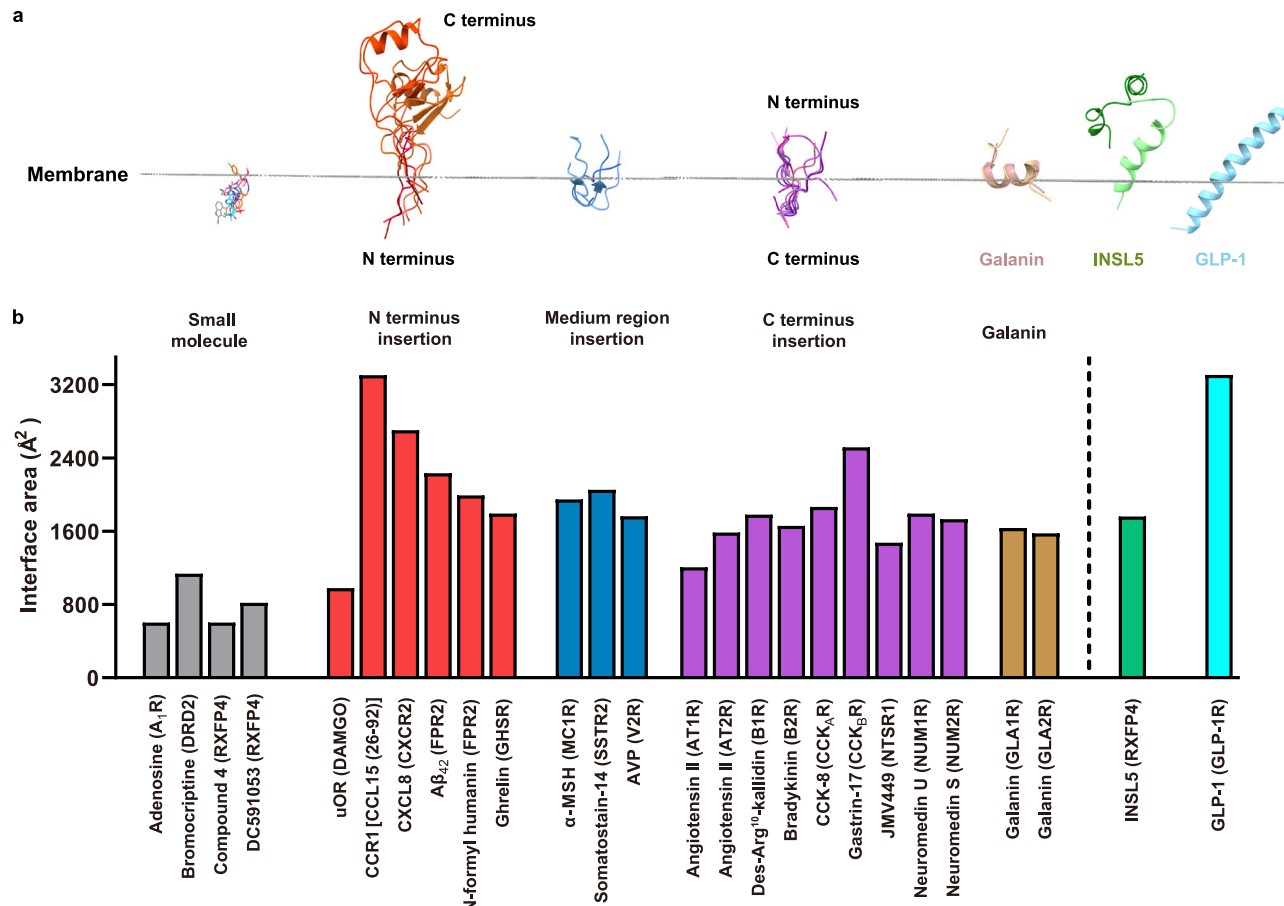

**Fig. 5 | Comparison of ligand binding modes. a** Conformations of different ligands during their insertion into the orthosteric pocket. The posture of INSL5 embedded in the membrane was compared with small molecules, galanin, GLP-1 and other peptides with N terminus, medium region and C terminus insertions into the TMD core. Receptor structures were omitted for clarity. **b** The interface area of peptide-receptor was measured with FreeSASA 2.0. Small molecules in grey; N termini insertions in red; medium region insertions in blue; C termini insertions in purple; galanin in brown; INSL5 in green and GLP-1 in cyan. Source data are provided as a Source Data file.

the cleft between the L1 domain and α-CT. Notably, IGF-1 is further inserted into a groove formed by L1 and CR domains (CRDs) of IGF-1R via its long C-domain loop. INSL5 that aligned to the insulin at site 1 eliminated interactions from the L1 domain-α-CT cleft and caused steric clashes with FnIII-1 and α-CT, respectively (Supplementary Fig. 12a, b). Similar phenomena were found when aligning INSL5 to IGF-1 bound by IGF-1R (Supplementary Fig. 12c, d). These observations reveal distinct ligand recognition mechanisms in the insulin superfamily and highlight that functional versatility is achieved by varying peptide sequences and ligand-binding pocket (Fig. 6e).

## Discussion

As one of the most important peptide-binding receptor subfamilies, RXFPs are promising drug targets for multiple diseases. In this study, we present three $G_i$-bound RXFP4 structures in complex with its endogenous ligand INSL5, RXFP3/RXFP4 dual agonist compound 4 and RXFP4-specific agonist DC591053. Because of the high flexibility and the relatively weak binding affinities, the INSL5 A chain and the morpholine ring of DC591053 showed low-resolution features compared with other regions of the ligands. Combined with mutagenesis, SAR analysis, and MD simulations, mechanisms of INSL5 recognition, peptidomimetic agonism, and subtype selectivity of RXFP4 were delineated, thereby expanding our understanding of the structural basis of functional versatility of the relaxin family peptide receptors.

The INSL5-bound RXFP4−G$_i$ complex structure presents a unique peptide-binding mode previously unknown and helps us elucidate an additional mechanism of activation related to peptide-binding class A GPCRs. Unlike the loop or "lay-flat" α-helix conformations adopted by other reported class A GPCR bound peptides, the B chain of INSL5 exhibits a single α-helix conformation that penetrates into the orthosteric pocket, while the A chain, similar to the extracellular domain (ECD) of class B1 GPCR, sits above the orthosteric pocket to interact the extracellular half of B chain as well as the extracellular surface of RXFP4. Despite variable receptor interaction modes, both A chain and B chain are indispensable to the functionality of INSL5, indicating the essence of such a peptide architecture in executing its action. This phenomenon has not been reported previously among peptidic ligands for GPCRs, but is a common feature (three intra-peptide disulfide bonds) of the insulin superfamily members.

High-resolution complex structures of compound 4- and DC591053-bound RXFP4 demonstrate both common and unique features of these two small molecule agonists in terms of peptidomimetic agonism and subtype selectivity. By structurally mimicking the C terminus residue W24$^B$, compound 4 and DC591053 occupy the bottom of the orthosteric pocket in a manner similar to INSL5 thereby displaying their peptidomimetic property. Meanwhile, the varying extents to which they contact RXFP4-specific residues form the foundation that governs receptor subtype selectivity, where DC591053 was discovered and validated as a RXFP4-specific agonist without observable cross-reactivity with RXFP3. Clearly, further structure-guided optimization of DC591053 towards better efficacy should be feasible with the support of the near-atomic level structural information.

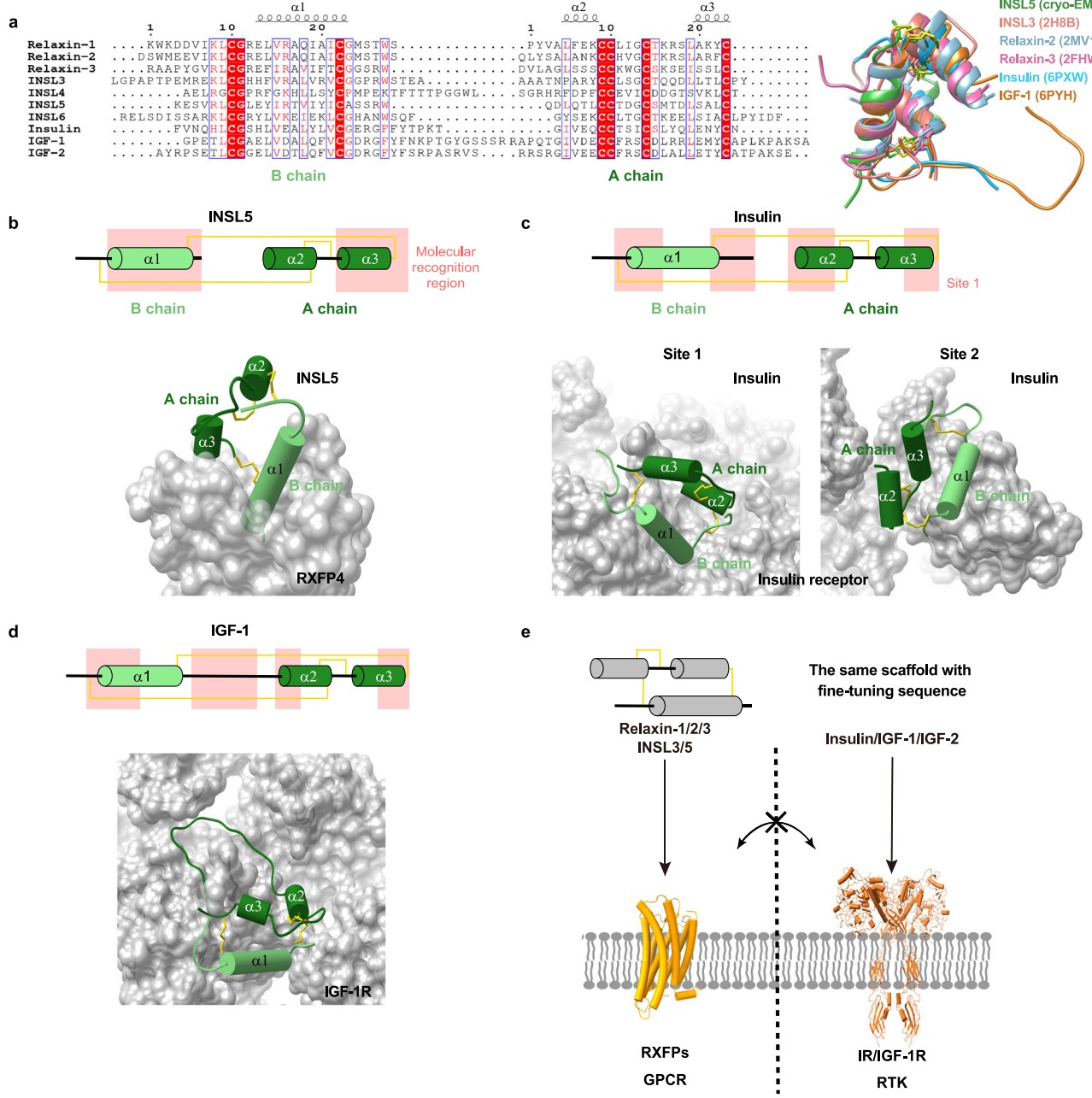

**Fig. 6 | Ligand recognition in the insulin superfamily. a** Sequence alignment (left) and structure superimposition (right) of peptides in the insulin superfamily colored as the labels. **b**–**d** Recognition of INSL5 (**b**) insulin (PDB code: 6PXW) (**c**) and IGF-1 (PDB code: 6PYH) (**d**) by cognate receptors, where the peptide (green) and receptor (grey) are shown in cartoon and surface, respectively. The peptide segments responsible for recognition are highlighted in the pink-shaded region and the three disulfide bonds in each peptide are indicated by yellow solid lines. **e** Distinct ligand recognition modes in the insulin superfamily.

Members of the insulin superfamily mediate a diverse array of signaling pathways through one TM or seven TMs receptors, representing an evolutionary lineage of functional versatility using a similar structural scaffold. To specifically activate corresponding receptors, two different and mutually exclusive peptide recognition modes (featured by α1 helix of INSL5 that inserts deeply to a buried pocket of RXFP4 and α2/α3 helixes of insulin/ IGF-1 that closely covers the planar interface of insulin receptor or IGF-1R) are employed, where variances in peptide sequence length and amino acid composition constitute the molecular basis of distinct functionalities. It appears that different regions of a peptide scaffold are able to interact with different types of receptors, conferring ligand specificity. In this manner, differences in signal transduction

between IR/IGF-1R (via homo- or hetero-dimerization) and GPCRs (via individual conformational alterations) are preserved to maximize functional versatility with a conserved peptide scaffold, especially for signal imitation and propagation. Unlike insulin and IGF-1 which mainly change the relative subdomain orientations to trigger downstream signaling, INSL5, as shown by the cryo-EM structure reported here, deeply inserts into the orthosteric pocket of RXFP4 (particularly the terminal residues R23[B] and W24[B] of the B chain) and induces conformational rearrangements of the ligand-binding pocket that further propagate to the intracellular side and render the outward movement of the intracellular half of TM6 as well as the G protein coupling. This information will greatly expand our knowledge on the signaling mechanisms of the insulin

superfamily and may advance the development of therapeutic agents for multiple diseases.

## Methods

### Construct

The full-length human RXFP4 (NCBI Reference Sequence: NM_181885.3) was cloned into a modified pFastBac vector (Invitrogen) with HA signal peptide to enhance receptor expression, followed by a 10× histidine tag and BRIL insertion at the N terminus. LgBiT subunit (Promega) was fused at the C terminus of RXFP4 connected by a 15-amino acid polypeptide linker. A dominant-negative human Gα$_{i2}$ (DNGi2) was generated by introducing S47N, G204A, E246A, and A327S substitutions in the Gα subunit as previously described[71]. The human Gβ1 with a C-terminal 15-amino acid polypeptide linker was followed by a HiBiT (peptide 86, Promega), and the scFv16 was modified with an N-terminal GP67 signaling peptide and a C-terminal 8× histidine tag. The engineered human Gα$_{i2}$, Gβ1, bovine Gγ2, and scFv16 were cloned into the pFastBac vector (Invitrogen), respectively. For cAMP accumulation assay, human RXFP4 and RXFP3 (NCBI Reference Sequence: NM_016568.3) were cloned into pCMV6 constructs (Ori-Gene Technologies). The mutant receptors were modified by site-directed mutagenesis in the setting of the WT constructs, with the primers designed by QuikChange Primer Design [QuickChange Primer Design (http://agilent.com.cn)] and carried out using Phanta Max Master (Vazyme). N-terminal Flag tag was added to both WT and mutant receptors for surface expression measurement. Sequences of all primers used in this study were provided in Supplementary Table 6, and all the constructs were confirmed by DNA sequencing.

### Production of INSL5 peptide

Recombinant INSL5 was designed to be produced from a single-chain INSL5 precursor in which the B chain (24 residues) and the A chain (21 residues) were connected by a specific C-peptide with the addition of a leader peptide at the N terminus. It was converted to two-chain human INSL5 by digesting with two proteinases after refolding (Supplementary Fig. 1a). Compared to the native hormone containing N-terminal pyroglutamate (pGlu, pE), the N-terminal glutamine (Gln, Q) of the recombinant INSL5 used in this study was not converted to pE (Supplementary Fig. 1b).

A gene encompassing the coding sequence of the INSL5 precursor (5′ end with Nde I recognition sequence and start codon, 3′ end with stop codon and Hind III recognition sequence) was designed and codon-optimized for high-level expression in *E. coli*. It was chemically synthesized and inserted into a pUC57 based vector (GenScript). The encoding DNA fragment of the INSL5 precursor was confirmed by DNA sequencing. The fragment of which was cleaved by Nde I and Hind III from the pUC57 plasmid and subsequently ligated into a pET vector that was pretreated with the same restriction enzymes using a T4-DNA polymerase. The expression construct was designated as pET-INSL5 plasmid and was transformed into competent *E. coli* cells derived from BL21 (DE3). After confirmation of the protein expression with IPTG induction, a single colony with a higher level was selected, cultured, and stored at −80 °C for future fermentation.

The above cells were cultivated in LB medium (ThermoFisher Scientific) at 37 °C and then inoculated for fermentation. At the end of fermentation, the biomass was harvested and the inclusion body was solubilized in 8 M urea solution and reduced by β-mercaptoethanol for 2 h. The reduced precursor was then refolded overnight, purified by chromatography, and cleaved with proteinases to generate the two-chain INSL5 with three pairs of correct disulfide bonds. After chromatographic purification, the mature two-chain INSL5 was analyzed by non-reducing SDS-PAGE and RP-HPLC (Supplementary Fig. 1).

The primary structure was confirmed by peptide mapping and 2-dimensional (2D) liquid chromatograph (LC)-MS. INSL5 was diluted 1:1 in the digestion buffer (100 mM Tris-HCl, 10 mM CaCl$_2$, pH 7.8) and

proteolytically cleaved with chymotrypsin (Sigma-Aldrich) at 37 °C for 1 h, with the mass ratio of enzyme to protein was 1:50. The separation of the peptides was performed with RP column (4.6 × 250 mm, 5 μm particle size, ThermoFisher Scientific). Eluents were A: water with 0.1% TFA; B: acetonitrile with 0.1% TFA. The elution gradient was as follows: 0 min, 10% B; 3 min, 10% B; 53 min, 60% B; 55 min, 100% B; 56 min, 100% B; and 60 min, 10% B at 30 °C with a flow rate of 0.4 mL/min. The eluted peptides were detected by UV absorbance at 230 nm. As for 2D LC-MS, the peptides were separated with Alliance HPLC (Waters) as the first dimension. Each peptide was cut individually and introduced to the second dimension with the Acquity UPLC (Waters) using another RP column (4.6 × 100 mm, 5 μm particle size, Halo), and was then detected by LTQ Orbitrap XL Mass Spectrometer (ThermoFisher Scientific). The following parameters were used for MS data acquisition: 100,000 resolution, scan range 150–2000 *m/z*, positive mode. Data analysis was conducted using the Qualbrowser application of Xcalibur software 2.1 (ThermoFisher Scientific) and ProMass Deconvolution 2.8 (Novatia). The amino acid sequences of chymotrypsin-generated peptides were assigned by matching molecular weight measured with theoretical sequence of a peptide using Expasy ProtParam tool (https://web.expasy.org/protparam/). The recombinant INSL5 peptide was subsequently verified for its bioactivity in CHO-K1 cells stably transfected with RXFP4 compared with an INSL5 standard (Phoenix Pharmaceuticals).

### Synthesis of DC591053

The RXFP4 agonist DC591053 was synthesized following procedures depicted in Supplementary Fig. 2a[72]. Commercially available 4-hydroxy-3-methoxybenzaldehyde (**1–1**) was treated with iodoethane to give **1–2**, which was refluxed in nitromethane to obtain **1–3** under the catalysis of ammonium acetate. Then compound **1–3** was reduced by LiAlH$_4$ to give key intermediate **1–4**. 5-Methoxy-1H-indole-3-carbaldehyde (**1–5**) was reacted with the wittig reagent methyl 2-(triphenyl-λ5-phosphanylidene)acetate (**1–6**) to give the corresponding α,β-unsaturated ester **1–7**, which was converted to the saturated ester **1–8** by catalytic hydrogenation. Hydrolysis of compound **1–8** afforded the key intermediate acid **1–9**. Amide **1–10** was generated by a coupling reaction of intermediates 2-(4-ethoxy-3-methoxyphenyl)ethan-1-amine (**1–4**) and 3-(5-methoxy-1H-indol-3-yl)propanoic acid (**1–9**). Then, amide **1–10** was treated with POCl$_3$ to afford the dihydroisoquinoline compound **1–11**. Asymmetric reduction with Noyori catalyst gave the *S*-isomer **1–12**, which was subjected to react with 4-morpholinecarbonyl chloride to provide the target product DC591053. It is a white solid characterized by $^1$H, $^{13}$C NMR and high-resolution mass spectra (HRMS) and determined to be 96.9% pure by column chromatography analyses ($^1$H NMR (500 MHz, DMSO-$d_6$) δ 10.58 (s, 1H), 7.21 (d, $J = 8.7$ Hz, 1H), 7.10 (d, $J = 2.0$ Hz, 1H), 6.93 (d, $J = 2.3$ Hz, 1H), 6.70 (dd, $J = 8.7$, 2.4 Hz, 1H), 6.64 (d, $J = 5.6$ Hz, 2H), 4.91 – 4.83 (m, 1H), 3.88 (q, $J = 7.0$ Hz, 2H), 3.74 (s, 3H), 3.69 (s, 3H), 3.57 (ddd, $J = 9.1$, 6.2, 2.8 Hz, 2H), 3.54 – 3.47 (m, 2H), 3.42 – 3.32 (m, 2H), 3.17 (ddd, $J = 12.5$, 6.1, 2.5 Hz, 2H), 3.03 (ddd, $J = 12.8$, 6.0, 2.6 Hz, 2H), 2.71 (ddd, $J = 27.7$, 13.5, 6.7 Hz, 3H), 2.64 – 2.56 (m, 1H), 2.07 (q, $J = 12.7$, 10.3 Hz, 2H), and 1.26 (t, $J = 7.0$ Hz, 3H). $^{13}$C NMR (125 MHz, DMSO-$d_6$) δ 163.40, 152.87, 147.47, 146.11, 131.51, 130.04, 127.33, 125.40, 122.99, 113.58, 112.01, 111.98, 111.71, 110.87, 100.12, 65.84, 63.71, 55.40, 55.31, 53.95, 47.40, 36.55, 27.53, 21.84, and 14.72. ESI-LRMS *m/z* 494.2 [M + H]$^+$. ESI-HRMS *m/z* calculated for C$_{28}$H$_{36}$N$_3$O$_5$ [M + H]$^+$ 494.2649, found 494.2650) (Supplementary Fig. 2b–d).

### Preparation of scFv16

ScFv16 was expressed in High-Five™ insect cells (ThermoFisher Scientific, Cat#B85502) as a secreted protein purified by Ni-sepharose chromatography column[49]. The HiLoad 16/600 Superdex 75 column (GE Healthcare) was used to separate the monomeric fractions of

scFv16 with a running buffer containing 20 mM HEPES and 100 mM NaCl, pH 7.4. The purified scFv16 was flash-frozen in liquid nitrogen with 10% glycerol and stored at −80 °C until use.

## Expression and purification of the RXFP4–$G_i$ complexes

Recombinant viruses of RXFP4, $G\alpha_{i2}$, $G\beta1$, and $G\gamma2$ were generated using Bac-to-Bac baculovirus expression system (Invitrogen) in *Spodoptera frugiperda* (*Sf*9) insect cells (Invitrogen, 10902-088). P0 viral stock was produced by transfecting 5 μg recombinant bacmids into *Sf*9 cells (2.5 mL, density of $1.5 \times 10^6$ cells per mL) for 96 h incubation and then used to produce high-titer P1 baculoviruses. High-Five™ insect cells were grown to a density of $3.2 \times 10^6$ cells per mL and infected with RXFP4, $G\alpha_{i2}$, $G\beta1$, and $G\gamma2$ P1 viral stocks at a ratio of 6: 1: 1: 1. The cells were cultured for 48 h at 27 °C after infection and harvested by centrifugation at $813 \times g$ for 20 min.

The cell pellets were lysed in buffer [20 mM HEPES, 100 mM NaCl and 100 μM TCEP, pH 7.4, supplemented with 10% (v/v) glycerol and EDTA-free protease inhibitor mixture (Bimake)], and the membrane was collected at $65,000 \times g$ for 30 min followed by homogenization in the same buffer. The formation of RXFP4–$G_i$ complexes was initiated by addition of 10 mM $MgCl_2$, 1 mM $MnCl_2$, 5 mM $CaCl_2$, 25 mU/mL apyrase (NEB), 15 μg/mL scFv16, ligands (20 μM INSL5, 50 μM compound 4 or 50 μM DC591053), 100 μM TCEP and 100 U salt active nuclease (Sigma-Aldrich) supplemented with protease inhibitor cocktail for 1.5 h incubation at room temperature (RT). The membrane was then solubilized with 0.5% (w/v) lauryl maltose neopentyl glycol (LMNG, Anatrace) and 0.1% (w/v) cholesterol hemisuccinate (CHS, Anatrace) with additional protease inhibitor cocktail for 3 h at 4 °C. The supernatant was isolated by centrifugation at $65,000 \times g$ for 1 h and incubated with Ni-NTA beads (GE Healthcare) for 1.5 h at 4 °C. The resin was collected and packed into a gravity flow column and washed with 10 column volumes of buffer A [20 mM HEPES, 100 mM NaCl, 5 mM $MgCl_2$, 1 mM $MnCl_2$ 100 μM TCEP, ligands (4 μM INSL5, 10 μM compound 4 or 10 μM DC591053), 0.1% (w/v) LMNG, 0.02% (w/v) CHS and 30 mM imidazole, pH 7.4], followed by washing with 20 column volumes of buffer B [essentially the same as buffer A with decreased concentrations of detergents 0.03% (w/v) LMNG, 0.01% (w/v) GDN and 0.008% (w/v) CHS containing 60 mM imidazole, pH 7.4]. The protein was eluted with five-column volumes of buffer C (buffer B with 300 mM imidazole, pH 7.4). The complexes were then concentrated using a 100-kD Amicon Ultra centrifugal filter (Millipore) and subjected to Superdex 200 10/300 GL column (GE Healthcare) with running buffer containing 20 mM HEPES, 100 mM NaCl, 100 μM TCEP, ligands (4 μM INSL5, 10 μM compound 4 or 10 μM DC591053), 0.00075% (w/v) LMNG, 0.00025% (w/v) GDN and 0.00025% (w/v) CHS, pH 7.4. The monomeric peak fractions were pooled and concentrated to 5–8 mg/mL.

## Cryo-EM data acquisition

The purified complex samples (3 μL at 5–8 mg/mL) were applied to glow-discharged holey grids (Quantifoil R1.2/1.3, 300 mesh) and subsequently vitrified using a Vitrobot Mark IV (ThermoFisher Scientific) set at 100% humidity and 4 °C. Cryo-EM images were acquired on a Titan Krios microscope (FEI) equipped with Gatan energy filter, K3 direct electron detector, and serial EM3.7. The microscope was operated at 300 kV accelerating voltage, at a nominal magnification of 46,685× in counting mode, corresponding to a pixel size of 1.071 Å. Totally, 9256 movies of the INSL5–RXFP4–$G_i$ complexes, 4639 movies of the compound 4–RXFP4–$G_i$ complexes, and 8230 movies of the DC591053–RXFP4–$G_i$ complexes were obtained, respectively, with a defocus range of −1.2 to −2.2 μm. An accumulated dose of 80 electrons per Å$^2$ was fractionated into a movie stack of 36 frames.

## Cryo-EM data processing

Dose-fractionated image stacks were subjected to beam-induced motion correction using MotionCor2.1. A sum of all frames, filtered according to the exposure dose, in each image stack was used for further processing. Contrast transfer function parameters for each micrograph were determined by Gctf v1.06. Particle selection, 2D, and 3D classifications were performed on a binned dataset with a pixel size of 2.142 Å using cryoSPARC v3.2.0 and RELION-3.1.1.

For the INSL5–RXFP4–$G_i$ complex, auto-picking yielded 10,618,534 particle projections that were subjected to two rounds of reference-free 2D classification to discard false-positive particles or particles categorized in poorly defined classes, producing 3,267,126 particle projections for further processing. This subset of particle projections was subjected to a round of maximum-likelihood-based 3D classification with a pixel size of 2.142 Å, resulting in one well-defined subset with 2,201,257 projections. Further 3D classification with a mask on the receptor produced one good subset accounting for 524,035 particles, which were then subjected to 3D refinement and Bayesian polishing with a pixel size of 1.071 Å. After the last round of refinement, the final map has an indicated global resolution of 3.19 Å at a Fourier shell correlation (FSC) of 0.143. Local resolution was determined using the Bsoft package (v2.0.3) with half maps as input maps.

For the compound 4–RXFP4–$G_i$ complex, auto-picking yielded 4,796,219 particle projections that were subjected to two rounds of reference-free 2D classification to discard false-positive particles or particles categorized in poorly defined classes, producing 787,382 particle projections for further processing. This subset of particle projections was subjected to a round of maximum-likelihood-based 3D classification with a pixel size of 2.142 Å, resulting in one well-defined subset with 469,428 projections. Further 3D classification with a mask on the receptor produced one good subset accounting for 243,800 particles, which were then subjected to 3D refinement and Bayesian polishing with a pixel size of 1.071 Å. The map with an indicated global resolution of 3.03 Å at a FSC of 0.143 was generated from the final 3D refinement. Local resolution was determined using the Bsoft package (v2.0.3) with half maps as input maps.

For the DC591053–RXFP4–$G_i$ complex, auto-picking yielded 8,996,005 particle projections that were subjected to two rounds of reference-free 2D classification to discard false-positive particles or particles categorized in poorly defined classes, producing 2,950,880 particle projections for further processing. This subset of particle projections was subjected to a round of maximum-likelihood-based 3D classification with a pixel size of 2.142 Å, resulting in one well-defined subset with 1,286,136 projections. Further 3D classification with a mask on the receptor produced one good subset accounting for 225,327 particles, which were then subjected to 3D refinement and Bayesian polishing with a pixel size of 1.071 Å. After the last round of refinement, the final map has an indicated global resolution of 2.75 Å at a FSC of 0.143. It was subsequently optimized using DeepEMhancer[73] before model building. Local resolution was determined using the Bsoft package (v2.0.3) with half maps as input maps.

## Model building and refinement

According to the expected quality of the resulting models using SWISS-MODEL (https://swissmodel.expasy.org/interactive) with the quality estimated by Global Model Quality Estimate (GMQE)[74], the cryo-EM structure of bradykinin–B2R complex (PDB code: 7F2O)[29] was used as the initial model of RXFP4 and scFv16, while the cryo-EM structure of $A_1R$–$G_i$ complex (PDB code: 6D9H)[71] was used to generate the initial model of G proteins. For the structure of compound 4–RXFP4–$G_i$ and DC591053–RXFP4–$G_i$ complexes, the coordinates of INSL5–RXFP4–$G_i$ complex were used as the starting point. Ligand coordinates and geometry restraints were generated using electronic Ligand Builder and Optimization Workbench (eLBOW)[75] and fitted to the cryo-EM density by LigandFit GUI[76] in PHENIX v1.18[77]. The model was docked into the EM density maps using UCSF Chimera v1.13.1[78], followed by iterative manual adjustment and rebuilding in COOT 0.9.4.1[79]. Real space refinement was performed using PHENIX v1.18[77].

The model statistics were validated using the module comprehensive validation (cryo-EM) in PHENIX v1.18[77,80]. Structural figures were prepared in UCSF Chimera v1.13.1, UCSF ChimeraX v1.0 and PyMOL v.2.1 (https://pymol.org/2/). The final refinement statistics are provided in Supplementary Table 1.

## Molecular dynamics simulation

MD simulations were performed by Gromacs 2020.1 (Supplementary Table 7). The INSL5–RXFP4 complexes were built based on the cryo-EM structure of the INSL5–RXFP4–$G_i$ complex and prepared by the Protein Preparation Wizard (Schrodinger 2017-4) with the G protein and scFv16 removed. The receptor chain termini were capped with acetyl and methylamide. All titratable residues were left in their dominant state at pH 7.0. To build MD simulation systems, the complexes were embedded in a bilayer composed of 237 POPC lipids and solvated with 0.15 M NaCl in explicit TIP3P waters using CHARMM-GUI Membrane Builder v3.5[81]. The CHARMM36-CAMP force filed[82] was adopted for protein, peptides, lipids and salt ions. The Particle Mesh Ewald (PME) method was used to treat all electrostatic interactions beyond a cut-off of 10 Å and the bonds involving hydrogen atoms were constrained using LINCS algorithm[83]. The complex system was first relaxed using the steepest descent energy minimization, followed by slow heating of the system to 310 K with restraints. The restraints were reduced gradually over 50 ns. Finally, restrain-free production run was carried out for each simulation, with a time step of 2 fs in the NPT ensemble at 310 K and 1 bar using the Nose-Hoover thermostat and the semi-isotropic Parrinello-Rahman barostat[84], respectively. The interface area was calculated by the program FreeSASA 2.0, using the Sharke-Rupley algorithm with a probe radius of 1.2 Å[85]. Similar simulation procedure and analysis were adopted for the MD simulations of INSL5 and its B chain, which were placed in a cubic box and the boundary of the box was at least 15 Å to the solute.

## Cell culture and transfection

CHO-K1 (ATCC, Cat#CCL-61) cells stably expressing human RXFP4 (hRXFP4-CHO) or RXFP3 (hRXFP3-CHO) were maintained in DMEM/F12 (Gibco) supplemented with 10% (v/v) fetal bovine serum (FBS) and 2 mM L-glutamine. Human embryonic kidney 293 T cells containing SV40 large T-antigen (HEK293T, ATCC, Cat#64127316) were maintained in DMEM (Gibco) supplemented with 10% (v/v) FBS, 1 mM sodium pyruvate (Gibco), 100 units/mL penicillin and 100 µg/mL streptomycin at 37 °C in 5% $CO_2$. For cAMP assays in mutants, HEK293T cells were seeded onto 6-well cell culture plates at a density of $7 \times 10^5$ cells per well. After overnight incubation, cells were transfected with WT or mutant receptors using Lipofectamine 3000 transfection reagent (Invitrogen). Following 24 h culturing, the transfected cells were ready for detection.

## Eu-labeled binding assay

CHO-K1 cells stably transfected with RXFP3 or RXFP4 were plated onto pre-coated poly-L-lysine 96-well plates. The competitive binding assays were performed with 5 nM Eu-H3 B1-22R (RXFP3) or Eu-R3/I5 (RXFP4) in the presence of increasing amounts of ligands as previously described[21,86,87]. Time-resolved fluorescence measurements were carried out at an excitation wavelength of 340 nm and an emission wavelength of 614 nm on a BMG POLARstar plate reader (BMG Labtech, Melbourne, Australia). Binding was performed in at least three independent experiments with triplicate determinations within each assay. Data are presented as means ± S.E.M. of specific binding and were fitted using a one-site binding curve in Prism software (GraphPad).

## cAMP accumulation assay

Inhibition of forskolin-induced cAMP accumulation by INSL5, compound 4, and DC591053 was measured by a LANCE Ultra cAMP kit (PerkinElmer). Ligands were verified for their bioactivity in the beginning in hRXFP4-CHO, which were ready for use after 24 h culturing. For assaying mutants, HEK293T cells were used 24 h post transfection. Cells were digested with 0.02% (w/v) EDTA and seeded onto 384-well microtiter plates at a density of $8 \times 10^5$ cells/mL in cAMP stimulation buffer [HBSS supplemented with 5 mM HEPES, 0.1% (w/v) bovine serum albumin (BSA) and 0.5 mM 3-isobutyl-1-methylxanthine]. The cells were stimulated with different concentrations of ligands plus 1.5 µM forskolin in RXFP4 and 4 µM forskolin in RXFP3. After 40 min incubation at RT, the Eu-cAMP tracer and ULight-anti-cAMP working solution were added to the plates separately to terminate the reaction followed by 60 min additional incubation. The time-resolved fluorescence resonance energy transfer (TR-FRET) signals were detected by an EnVision multilabel plate reader (PerkinElmer) with the emission window ratio of 665 nm over 620 nm under 320 nm excitation. Data were normalized to the maximal response of WT receptor.

## Receptor surface expression

Cell membrane expression was determined by flow cytometry to detect the N-terminal Flag tag on the WT and mutant receptor constructs transiently expressed in HEK293T cells. Briefly, approximately $2 \times 10^5$ cells were blocked with PBS containing 5% BSA (w/v) at RT for 15 min, and then incubated with 1:300 anti-Flag primary antibody (diluted with PBS containing 5% BSA, Sigma-Aldrich, Cat#F3165, purified IgG1 subclass) at RT for 60 min. The cells were then washed three times with PBS containing 1% BSA (w/v) followed by 60 min incubation with 1:1000 anti-mouse Alexa Fluor 488 conjugated secondary antibody (diluted with PBS containing 5% BSA, Invitrogen, Cat#A-21202) at RT in the dark. After washing three times, cells were resuspended in 200 µL PBS containing 1% BSA for detection by NovoExpress 1.2.1 (Agilent) utilizing laser excitation and emission wavelengths of 488 nm and 530 nm, respectively. For each sample, 20,000 cellular events were collected, and the total fluorescence intensity of the positive expression cell population was calculated. The gating strategy was shown in Supplementary Fig. 13. Data were normalized to the WT receptor and parental HEK293T cells.

## Statistical analysis

All functional study data were analyzed using GraphPad Prism 8.3 (GraphPad Software) and presented as means ± S.E.M. from at least three independent experiments. Dose-response curves were evaluated with a three-parameter logistic equation. The significance was determined with either a two-tailed Student's *t*-test or one-way ANOVA with Dunnett's multiple comparison test, and $P < 0.05$ was considered statistically significant.

## Reporting summary

Further information on research design is available in the Nature Portfolio Reporting Summary linked to this article.

# Data availability

The data that support this study are available from the corresponding authors upon reasonable request. The cryo-EM density maps have been deposited in the Electron Microscopy Data Bank (EMDB) under accession codes EMD-33871 (INSL5–RXFP4–$G_i$ complex), EMD-33888 (compound 4–RXFP4–$G_i$ complex), and EMD-33889 (DC591053–RXFP4–$G_i$ complex). Coordinates have been deposited in the Protein Data Bank (PDB) under accession codes 7YJ4 (INSL5–RXFP4–$G_i$ complex), 7YK6 (compound 4–RXFP4–$G_i$ complex), and 7YK7 (DC591053–RXFP4–$G_i$ complex). The data underlying Figs. 1e, 2f–g, 3c, 3e, 3h, 5b, Supplementary Figs. 1c, 1f–g, 2e–g, 3b–d, 4a–c, 10b–g and Supplementary Tables 3, 4 and 5 are provided as a Source Data file. Source data are provided with this paper.

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

## Acknowledgements

We are grateful to Jiao Yu, Tania Ferraro, and Sharon Layfield for their technical assistance. This work was supported by the National Natural Science Foundation of China 81872915 (M.-W.W.), 82073904 (M.-W.W.), 82121005 (D.Y.), 81973373 (D.Y.), 82130105 (H.L.) and 21704064 (Q.T.Z.); National Science & Technology Major Project of China–Key New Drug Creation and Manufacturing Program 2018ZX09735–001 (M.-W.W.) and 2018ZX09711002–002–005 (D.Y.); STI2030-Major Project 2021ZD0203400 (Q.T.Z.); the National Key Basic Research Program of China 2018YFA0507000 (M.-W.W.); Hainan Provincial Major Science and Technology Project ZDKJ2021028 (D.Y. and Q.T.Z.) and Shanghai Municipality Science and Technology Development Fund 21JC1401600 (D.Y.), the Victorian Government's Operational Infrastructure Support Program (R.A.D.B.) and National Health and Medical Research Council of Australia Research Fellowship 1135837 (R.A.D.B.). The cryo-EM data were collected at the Cryo-Electron Microscopy Research Center, Shanghai Institute of Materia Medica, Chinese Academy of Sciences.

## Author contributions

Y.C. designed expression constructs, purified the receptor complexes, screened the specimen, prepared the final samples for cryo-EM data collection, conducted map calculation, built the models of the

complexes, performed signaling experiments, and participated in manuscript preparation; Q.T.Z. performed model building, structural analysis, MD simulations, and figure preparation and participated in manuscript writing; J.W. synthesized DC591053 with the guidance of H.L.; Y.W.X. performed structure refinement and model building under the supervision of H.E.X.; Y.W. and Q.Z. produced recombinant INSL5 under the supervision of C.S.; J.H.Y., F.H.Z., C.-W.C., and X.Q.C. took part in method development and functional experiments; Y.B.W. and C.H.L assisted in the synthesis of DC591053; R.A.D.B. organized ligand binding assay; D.H.Y., H.L., and M.-W.W. initiated the project, supervised the studies, analyzed the data, and wrote the manuscript with inputs from all co-authors.

## Competing interests

The authors declare no competing interests.
