## [Peer Review File · Nature Communications]

Ligand recognition mechanism of the human relaxin family peptide receptor 4 (RXFP4)Reviewers' Comments:

Reviewer #1:

Remarks to the Author:

Chen et al., reported three cryo-EM structures of RXFP4-Gi complex bound to an endogenous ligand INSL5 and two compounds with distinct subtype selectivity. The paper nicely describes how the insulin-like peptide in the insulin superfamily is recognized by the G protein-coupled receptor, and provides insights into mechanism of subtype selectivity. The structures description is stated satisfactorily, and the data generally support the conclusions. However, revision should be made before I can recommend its publication.

Major points:

1. Relaxin-3 can activate both RXFP3 and RXFP4, while INSL5 selectively activates RXFP4, and is a low-affinity antagonist of RXFP3. The authors should discuss how the selectivity is achieved, and functional validation is encouraged. Comparison of structures of RXFP4 in inactive and active states should provide insights into mechanism of receptor activation by INSL5, which may explain the antagonism of INSL5 for RXFP3. The inactive structure of RXFP4 could be made via homology modeling since no experimental model is available.
2. Experimental details of screening for discovering the lead compound DS591053 and INSL5 refolding should be provided.
3. It seems that the EM densities for some regions of INSL5 and DC591053 are poor in the Supplemental figure 5. I recommend the authors can optimize the density of the receptor and ligands by local refinement. If EM densities are not good enough to unambiguously model the ligands, the authors should discuss the limitation of structures in the text. The density for ligands should be moved to the main figure.
4. The authors should analyze the expression level of RXFP4 mutants, which can affect the interpretation of the results especially for those with loss of response.

Minor points:

1. The typos and grammar errors should be corrected throughout the paper.
 - Lane 32 and 209: "orthostatic: should be "orthosteric".
 - Lane 219: replace "significant challenge" by "very challenging"
 - Lane 51 "with a relatively short N-terminal tail rather than LRR"
 - Lane 226: "developed demonstrating".....
2. The authors should mention why developing selective agonists for RXFP4 is very important in the introduction. Is this therapeutically relevant?
3. The paragraph "Characterization of DC591053" can be moved to the method section.
4. Some characters in main and Supplementary Figures are too small. The BW numbers in Figs are barely visible.

Reviewer #2:

Remarks to the Author:

The manuscript presents a description for the interactions between RXFP4-Gi and the INSL5 peptide in the orthosteric pocket, and for peptidomimetic agonists such as the compound 4 (C4) and the

synthesized DC591053. Results from 3D cryo-electron microscopy structures, resolved at 3.19 to 2.75 Å resolution, are discussed in the context other GPCR-peptide complexes. In addition, functional in-vitro assays for the receptor construct and recombinant INSL5 were performed to account for the receptor expression, activation in response to endogenous or synthetic agonists, and ligand binding. Results suggest a specific binding mode of the INSL5 peptide in the receptor bundle in comparison to other insulin-like peptides such as GLP1, relaxins1-3, or IGF1-2. The hydrophobic pocket hosting the endogenous hormone or the peptidomimetic molecules share similar features of stacking of aromatic rings involving W97, F105, Y121, F291, H299 of the receptor and the indole group of the ligands. The similarities in the binding mode among the RXFP4-Gi complexes suggest that selectivity may be enhanced by designing molecules showing favorable stacking of aromatic rings which may be part of the activation mechanisms involved in this subfamily of receptors.

General Concerns

Overall results from the experimental procedures show evidence of the interactions of the INSL5 ligand and synthetic molecules, compound 4 and DC591053. However, the computational procedures require more clarification regarding the strategy and algorithms implemented. For example:

1. Docking a known structure on a new cryo-EM map could be advantageous for a first approximation on solving unknown structures. In the model building and refinement section authors mention the structures used as initial model for solving RXFP4 and the G-protein (lines 521-522). Could authors explain why they chose such structure for implementation? Are those structures close related to the RXFP4? Is there any justification in terms of the structure, sequence similarity, structure resolution or any other comparable feature? Could the authors propose a strategy for designing a robust methodology for solving membrane proteins from cry-EM data in case no similar structures are available?
2. In line 524, could the authors provide more details for using restraints on the ligand coordinates? Could authors explain how the ligands coordinates were assigned? What kind of restraints were imposed, positional, dihedral, distances, etc.?
3. In line 526, What does it mean manual adjustment of the model and rebuilding? Is there any mathematical algorithm involved in the structure fitting in the density map?
4. What algorithm is implemented in the real space refinement in the PHENIX software?
5. For the structures C4 and DC591053 bound to RXFP4-Gi, the initial structure of the complex INSL5-RXFP4-Gi was used (line 523). May the authors clarify the modeling process of compound 4 and DC591053, and how the atom positions were assigned for the synthetic agonists?

In the molecular dynamics simulation section authors describe a setup for the INSL5-RXFP4 complex in a POPC lipid bilayer.

1. May the authors provide more context on the purpose of the MD simulation study? Was it designed to test the consistency of the interactions found in the resolved structure against those found in a MD all-atom Force Field calculation?
2. From figures 8 and 9 in Suppl-info, there is a shift of the INSL5 in the receptor binding pocket. As the receptor binding pocket seems well preserved (Suppl Figure 8c) What interactions produce the C-terminus motion of W24? Did authors consider water mediated interactions or hydration level in the interhelical region? Reports of internal water molecules in crystal structures in GPCRs are known in literature.
3. Authors may include more details of the simulation protocol for reproducibility: I. system setup, II. equilibration and production protocols III analysis for data collection. Calculation of the binding free energy could be relevant to demonstrate selectivity of DC591053 in the RXFP4-Gi complex. Indeed, in line 183 of the main text authors mention that "C-terminal α -helix of the B-chain could stably insert into the orthosteric pocket through its tip residues". This statement is misleading as the starting configuration included already the INSL5 in the binding pocket. A description of the insertion mechanism would require the calculation of the transition from unbound to bound states using, for example, umbrella sampling.

In section Expression and purification of the RXFP4-Gi complex describes the procedure to produce the recombinant receptors in insect cells. May the authors briefly describe the procedure to produce the receptor mutants?

For structure validation it is recommended to report the Rama-Z score (Structure 28, 1249–1258.e1–e2, November 3, 2020), which provides a criterion for improbable backbone geometry $|Z| > 3$, $2 < |Z| < 3$ possible geometry, and $|Z| < 2$ for normal geometry. In Suppl-Table 1, the favorable, allowed and outliers was reported as a percentage, which is not a definitive criterion for a good shape of the Ramachandran angles distribution.

On lines 51-52 authors describe differences in the N-terminal tail, between RXFP1-2 and RXFP3-4, and mention 43% of sequence identity. Could authors clarify whether the sequence identity refers to the TM domains, the N-terminal, or was it for the overall structure?

POINT-BY-POINT RESPONSES TO THE REVIEWERS' COMMENTS

Reviewer #1 (Remarks to the Author):

Chen et al., reported three cryo-EM structures of RXFP4-Gi complex bound to an endogenous ligand INSL5 and two compounds with distinct subtype selectivity. The paper nicely describes how the insulin-like peptide in the insulin superfamily is recognized by the G protein-coupled receptor, and provides insights into mechanism of subtype selectivity. The structures description is stated satisfactorily, and the data generally support the conclusions. However, revision should be made before I can recommend its publication.

Major points:

1. Relaxin-3 can activate both RXFP3 and RXFP4, while INSL5 selectively activates RXFP4, and is a low-affinity antagonist of RXFP3. The authors should discuss how the selectivity is achieved, and functional validation is encouraged. Comparison of structures of RXFP4 in inactive and active states should provide insights into mechanism of receptor activation by INSL5, which may explain the antagonism of INSL5 for RXFP3. The inactive structure of RXFP4 could be made via homology modeling since no experimental model is available.

Response: That is an excellent question. Previous findings have identified differentiated requirements of the peptide B chain C-terminal conformation for efficient activation between RXFP4 and RXFP3, consistent with their distinct structural properties seen in the NMR studies (Figure X1, PMID: 27404393). Different from RXFP4 that is largely tolerant to flexible or rigid B chain C terminus, RXFP3 strictly requires flexible B chain C terminus. Thus, a rigid B chain C terminus in INSL5 might be one of the reasons for its incapability to activate RXFP3 (PMID: 27404393).

Figure X1. Sequence alignment and structure comparison of INSL5 and relaxin-3 in previous research (PMID: 27404393). **a**, Amino acid sequence alignment of human INSL5 and relaxin-3 B chains. The residues exchanged from relaxin-3 to INSL5 in the present work are shown in red. **b**, The previously reported solution structures of INSL5 and relaxin-3. The residues exchanged from relaxin-3 to INSL5 are shown as red sticks and labelled. The length of B chain C-terminal α -helix of INSL5 is longer than that of relaxin-3, mainly because of the adjacent two glycines (-CGGSRW) in relaxin-3 that is absent in INSL5 (-CASSRW).

To further address this point, we performed additional alanine mutation and amino acid switching experiments in the equivalent positions of TMs 3, 5 and 7 between RXFP4 and RXFP3 (around the orthosteric binding pocket). As shown in Figure X2, INSL5 was totally inactive in RXFP4 single mutants L118^{3.29}S and L118^{3.29}A as well as double mutants L118^{3.29}S+V122^{3.33}S and L118^{3.29}A+V122^{3.33}A, where relaxin-3 retained partial activity although the curves shifted to the right (by 3.2-fold, 5.4-fold, 14.8-fold and 9.7-fold, respectively). For comparison, relaxin-3 activated S159^{3.29}A, S159^{3.29}L, S159^{3.29}L+S163^{3.33}V and S159^{3.29}A+S163^{3.33}A in RXFP3 albeit with reduced potencies. T295^{7.39}V did not destroy the response of RXFP4 to INSL5 and relaxin-3, but V375^{7.39}T in RXFP3 both impaired the potency (by 6.1-fold) and E_{max} (66.5% of the wild-type, WT) of relaxin-3 (INSL5 was inactive in WT and all the four RXFP3 mutants). Therefore, S159^{3.29}, S163^{3.33} and V375^{7.39} in RXFP3 and L118^{3.29}, V122^{3.33} and T295^{7.39} in RXFP4 are likely involved in RXFP3 vs. RXFP4 subtype selectivity, consistent with the observations in RXFP3/RXFP4 chimeric receptor studies (PMID: 18582868). Clearly, it will be helpful to further elucidate such a selectivity when a cryo-EM structure of RXFP3 is available.

Figure X2. Key residues likely involved in receptor subtype selectivity. **a**, Binding mode of INSL5 (green) with RXFP4 (orange) in the cryo-EM structure. L118^{3,29}, V122^{3,33} and T295^{7,39} (marked red) probably contribute to INSL5 and relaxin-3 selectivity between RXFP4 and RXFP3. **b-c**, Effects of INSL5 and relaxin-3 on cAMP accumulation in wild-type (WT) and mutant RXFP4. **d**, Effects of relaxin-3 on cAMP accumulation in WT and mutant RXFP3. Data are shown as means \pm S.E.M. of at least three independent experiments. max, maximum response.

2. Experimental details of screening for discovering the lead compound DS591053 and INSL5 refolding should be provided.

Response: Thanks for the comment. We screened our in-house tetrahydroisoquinoline library to discover novel RXFP4 agonists using the cAMP accumulation assay. As a result, six compounds were found to display potent RXFP4 agonist activities (Figure X3) with DC591053 being the best ($pEC_{50} = 7.24 \pm 0.12$, $n = 3$ as measured in stably-transfected CHO-K1 cells). Importantly, DC591053 neither reacted with RXFP3 nor parental CHO-K1 cells. The screening studies will be summarized in another manuscript currently in preparation. To reflect this, we have revised the manuscript: “We screened our in-house tetrahydroisoquinoline library aimed at discovering novel RXFP4 agonists using cAMP accumulation assay. **Of the six compounds displaying RXFP4 agonist activities (data not shown), the lead compound, DC591053 ((S)-(7-ethoxy-6-methoxy-1-(2-(5-methoxy-1H-indol-3-yl)ethyl)-3,4-dihydroisoquinolin-2(1H)-yl)(morpholino)methanone), exhibited the best agonism. It was identified and synthesized from the commercially available compound 4-hydroxy-3-methoxybenzaldehyde, followed by alkylation reaction, reduction, Wittig reaction, cyclization, asymmetric reduction reaction, and condensation reaction (Supplementary Fig. 2a-d).**”

As far as the experimental details of INSL5 refolding, the relevant part of the manuscript has been expanded as: “The above cells were cultivated in LB medium (ThermoFisher Scientific) at 37°C and then inoculated for fermentation. At the end of fermentation, the biomass was harvested and the inclusion body was ~~recovered for refolding~~ **solubilized in 8 M urea solution and reduced by β -mercaptoethanol for 2 h. The reduced precursor was then refolded overnight.** ~~The refolded precursor was~~ purified by chromatography and cleaved with proteinases to generate the two-chain INSL5 with three pairs of correct disulfide bonds.”

Figure X3. Inhibition of forskolin-stimulated cAMP accumulation by ten representative compounds screened in CHO-K1 cells overexpressing hRXFP4. Each compound was tested in quadruplicate and the experiment was repeated three times. Agonist activity was expressed as percentage of the maximum response (max) to INSL5 in hRXFP4-CHO-K1 cells. Normalized values were plotted vs. ligand concentration using GraphPad Prism 8 and expressed as means \pm S.E.M.

3. It seems that the EM densities for some regions of INSL5 and DC591053 are poor in the Supplemental figure 5. I recommend the authors can optimize the density of the receptor and ligands by local refinement. If EM densities are not good enough to unambiguously model the ligands, the authors should discuss the limitation of structures in the text. The density for ligands should be moved to the main figure.

Response: We thank the reviewer for the valuable suggestion. To further optimize the density of the receptor and ligands, we rerun particle picking and tried several rounds of local refinements with different parameters as well as DeepEMhancer (PMID: 34267316), but failed to improve. The poor electron densities of the INSL5 A chain and morpholine ring of DC591053 were probably due to their relative flexibilities and weak binding affinities. As the reviewer suggested, the near-atomic resolution models of the three ligands in the cryo-EM density maps have now been moved to Figure 1. Meanwhile, the method part was expanded accordingly: “After the last round of refinement, the final map has an indicated global resolution of 2.75 Å at a FSC of 0.143. It was subsequently optimized using DeepEMhancer⁷³ before model building.”

To reflect the limitation of our structures, the following statement has been added to the discussion: “In this study, we present three G_i-bound RXFP4 structures in complex with its endogenous ligand INSL5, RXFP3/RXFP4 dual agonist compound 4 and RXFP4-specific agonist DC591053. Because of the high flexibility and the relatively weak binding affinities, the INSL5 A chain and morpholine ring of DC591053 showed low-resolution features compared with other regions of the ligands.”

4. The authors should analyze the expression level of RXFP4 mutants, which can affect the interpretation of the results especially for those with loss of response.

Response: We thank the reviewer for the suggestion. Detailed information of the mutants and their surface expression levels have been included in the revised Supplementary Table 4 (Table X1 below). As mentioned in the manuscript, the RXFP4 mutants T121A and H299A displayed significantly lower response to the three ligands, but their expression levels were between 40.11% and 84.87% of the WT. Three RXFP4 mutants (R208A, W97A and E100A) could be activated by at least one of the three ligands, thus the loss of response to other two ligands appears not entirely associated with surface expression.

Table X1. Expression level of wild-type and mutant receptors.

Receptor	Mutation	Cell surface expression (% WT)
RXFP4	WT	100

	W97A	59.76 ± 0.21****
	E100A	81.83 ± 4.56
	D104A	38.82 ± 1.34****
	F105A	69.26 ± 2.69**
	T121A	40.11 ± 2.25****
	R194A	29.81 ± 1.01****
	Q205A	80.26 ± 2.04
	R208A	27.07 ± 1.09****
	K273A	35.92 ± 2.69****
	W279A	151.25 ± 6.66****
	Y284A	123.5 ± 8.30**
	H299A	84.87 ± 8.49
	L118S+V122S	24.79 ± 2.68****
	Q205H	91.09 ± 1.43
	R208K	68.61 ± 4.73****
	T295V	66.15 ± 7.26****
RXFP3	WT	100
	S159L+S163V	78.75 ± 2.54***
	H268Q	96.03 ± 2.79
	K271R	97.86 ± 1.15
	V375T	95.63 ± 3.30

Cell surface expression was measured by flow cytometry. Values were normalized to the wild-type in HEK293T cells. Data shown are means ± S.E.M. of at least three independent experiments. One-way ANOVA was used to determine statistical difference (*P< 0.05, **P< 0.01, ***P< 0.001, ****P< 0.0001).

Minor points:

1. *The typos and grammar errors should be corrected throughout the paper.*

-Lane 32 and 209: “orthostatic: should be “orthosteric”.

Response: These points are well taken, thanks.

-Lane 219: replace “significant challenge” by “very challenging”

Response: This point is well taken. We have revised the relevant sentence as: “Since the sequence identity of the ligand-binding pocket between RXFP3 and RXFP4 is 86.36%, development of receptor subtype-selective ligands is ~~significant challenge~~ **very challenging**.”

-Lane 51 “with a relatively short N-terminal tail rather than LRR”

Response: This point is well taken. We have revised the relevant sentence as: “RXFP3 and RXFP4 have distinct binding properties with ~~a~~ **relatively short N-terminal tails** rather than LRR.”

-Lane 226: “developed demonstrating”

Response: This point is well taken. We have revised the relevant sentence as: “To overcome this hurdle, DC591053 was developed ~~to demonstrating~~ **a full agonism** at RXFP4 ($pEC_{50} = 7.24 \pm 0.12$) without observable cross-reactivity with RXFP3 (**Fig. 3e**).”

2. *The authors should mention why developing selective agonists for RXFP4 is very important in the introduction. Is this therapeutically relevant?*

Response: We thank the reviewer for the valuable comments. *In vivo*, the overlapping expression pattern between RXFP4 and RXFP3 (PMIDs: 36184065; 27774604) as well as the related physiological properties following their activation were reported, including the influences on food intake, body weight, energy rebalance and feeding behavior. However, the precise roles of RXFP3 and RXFP4 in these processes are still unclear, because most available ligands all have *in vitro* cross-reactivity between RXFP3 and RXFP4. Thus, a subtype specific agonist will be helpful to distinguish these two receptor subtypes. We have added the following statements to the introduction: “In addition to peptidic analogues, small molecule modulators have been reported in recent years. Compound 4, an amidino hydrazone-based scaffold identified by Novartis, is an RXFP3/RXFP4 dual agonist¹⁸. **Because high cross-reactivity, it cannot be used therapeutically. *In vivo*, the overlapping expression pattern between RXFP4 and RXFP3 as well as their distinct physiological properties^{19,20} call for subtype specific agonists which will likely be valuable to different clinical applications.** However, selective RXFP4 agonists discovered via high-throughput screening campaigns and follow-up structural modifications displayed deficiencies in solubility, potency and toxicity^{21,22}.”

3. The paragraph “Characterization of DC591053” can be moved to the method section.

Response: Thanks for the suggestion. We have significantly revised the manuscript by moving some chemistry details to the method section.

4. Some characters in main and Supplementary Figures are too small. The BW numbers in Figs are barely visible.

Response: This point is well taken and all related figures have been revised accordingly.

Reviewer #2 (Remarks to the Author):

The manuscript presents a description for the interactions between RXFP4-Gi and the INSL5 peptide in the orthosteric pocket, and for peptidomimetic agonists such as the compound 4 (C4) and the synthesized DC591053. Results from 3D cryo-electron microscopy structures, resolved at 3.19 to 2.75 Å resolution, are discussed in the context other GPCR-peptide complexes. In addition, functional in-vitro assays for the receptor construct and recombinant INSL5 were performed to account for the receptor expression, activation in response to endogenous or synthetic agonists, and ligand binding. Results suggest a specific binding mode of the INSL5 peptide in the receptor bundle in comparison to other insulin-like peptides such as GLP1, relaxins1-3, or IGF1-2. The hydrophobic pocket hosting the endogenous hormone or the peptidomimetic molecules share similar features of stacking of aromatic rings involving W97, F105, Y121, F291, H299 of the receptor and the indole group of the ligands. The similarities in the binding mode among the RXFP4-Gi complexes suggest that selectivity may be enhanced by designing molecules showing favorable stacking of aromatic rings which may be part of the activation mechanisms involved in this subfamily of receptors.

General Concerns

Overall results from the experimental procedures show evidence of the interactions of the INSL5 ligand and synthetic molecules, compound 4 and DC591053. However, the computational procedures require more clarification regarding the strategy and algorithms implemented. For example:

1. Docking a known structure on a new cryo-EM map could be advantageous for a first approximation on solving unknown structures. In the model building and refinement section authors mention the structures used as initial model for solving RXFP4 and the G-protein (lines 521-522). Could authors explain why they chose such structure for implementation? Are those structures close related to the RXFP4? Is there any justification in terms of the structure, sequence similarity, structure resolution or any other comparable feature? Could the authors propose a strategy for designing a robust methodology for solving membrane proteins from cry-EM data in case no similar structures are available?

Response: Thanks for the question. The initial model of fully active RXFP4 was built via one well-known webserver, SWISS-MODEL (PMID: 29788355, <https://swissmodel.expasy.org/interactive>), where two database search methods (BLAST and HHblits) were adopted. The top 10 structural templates ranked according to expected quality of the resulting models, as estimated by Global Model Quality Estimate (GMQE), were shown in Table X2. By analysis and comparison

of them in terms of GMQE, the sequence identity to the target and experimental method used to obtain the structure (cryo-EM structures of G protein-bound fully active receptor conformation templates are preferred), the cryo-EM structure of the type 2 bradykinin receptor in complex with the bradykinin (PDB code: 7F2O) was chosen as the initial model template of RXFP4 with the highest GMQE score (0.56) and good sequence identity (25.94%). As far as G protein is concerned, we used the G protein construct identical to previously described the A₁R–G_i cryo-EM structure (PDB code: 6D9H), for model building. The corresponding sentence in the manuscript has been revised: “According to the expected quality of the resulting models using SWISS-MODEL (<https://swissmodel.expasy.org/interactive>) with the quality estimated by Global Model Quality Estimate (GMQE)⁷⁴, the cryo-EM structure of bradykinin–B2R complex (PDB code: 7F2O)²⁹ was used as the initial model of RXFP4 and scFv16, while the cryo-EM structure of A₁R–G_i complex (PDB code: 6D9H)⁷¹ was used to generate the initial model of G proteins.”

The methodology development toward robust solving membrane protein structure model from cryo-EM data without reference structure is one of the fundamental and important tasks for computational biologists. Impressively, machine-learning technology has joined this effort which brings many promising tools such as DeepTracer (PMID: 33361332), CryoDRGN (PMID: 33542510) and SAUA-FFR (PMID: 34142833).

Table X2. The top 10 structure templates for the RXFP4 homology model identified by the SWISS-MODEL webserver (<https://swissmodel.expasy.org/interactive>), ranked by the GMQE score.

PDB code	Receptor name	GMQE	Sequence similarity (%)	Sequence identity (%)	Database search method	Experimental method
7F2O	B2 bradykinin receptor	0.56	34%	25.94%	HHblits	Cryo-EM
6OS1	Type-1 angiotensin II receptor	0.55	35%	26.74%	HHblits	X-ray
6JOD	Type-2 angiotensin II receptor	0.54	35%	30.21%	HHblits	X-ray
7SK8	Atypical chemokine receptor 3	0.54	35%	26.28%	HHblits	Cryo-EM
7SK4	Atypical chemokine receptor 3	0.54	34%	26.28%	HHblits	Cryo-EM
7SBF	Mu-type opioid receptor	0.54	35%	29.41%	BLAST	Cryo-EM
7SK7	Atypical chemokine receptor 3	0.54	34%	26.28%	HHblits	Cryo-EM
6DO1	Type-1 angiotensin II receptor	0.54	35%	26.74%	HHblits	X-ray
7SK3	Atypical chemokine receptor 3	0.54	34%	26.28%	HHblits	Cryo-EM
7WVW	N-formyl peptide receptor 2	0.54	35%	29.55%	HHblits	Cryo-EM

2. In line 524, could the authors provide more details for using restraints on the ligand coordinates? Could authors explain how the ligands coordinates were assigned? What kind of restraints were imposed, positional, dihedral, distances, etc.?

Response: Thanks for the comments. Both the ligand structures and restraints were generated by the electronic Ligand Builder and Optimization Workbench (eLBOW, PMID: 19770504) implanted in PHENIX v1.18 with the input of the chemical information of the desired ligands (https://phenix-online.org/documentation/reference/elbow_gui.html). Then, the ligand coordinates were fitted to the cryo-EM density by LigandFit GUI (PMID: 16855309) and incorporated into the protein to be refined by PHENIX and COOT. The ligand restraint files (in CIF format) generated by eLBOW have been uploaded to the submission system as Ligand_cif.zip (including Compound4.cif and DC591053.cif). Both references and the main text have been revised to reflect this point: “Ligand coordinates and geometry restraints were generated using electronic Ligand Builder and Optimization Workbench (eLBOW)⁷⁵ and fitted to the cryo-EM density by LigandFit GUI⁷⁶ phenix.elbow in PHENIX v1.18⁷⁷.”

3. In line 526, What does it mean manual adjustment of the model and rebuilding? Is there any mathematical algorithm involved in the structure fitting in the density map?

Response: Thanks for the comment. In the model building process, the initial template was rigidly fitted to the electron density maps using local optimization algorithm in UCSF Chimera v1.13.1. Then, based on electron density, the manual

adjustment of model and rebuilding were performed for these residues of poor density or geometry in COOT 0.9.4.1 (PMIDs: 20383002 and 15572765). Such actions were taken primarily by means of the real-space refinement engine, which handles the refinement of the atomic model against an electron-density map and the regularization of the atomic model against geometric restraint. Based on the comparison of a model against electron density and comprehensive geometrical checks for protein structures from validation tools, we could further optimize the model interactively through other tools including “Regularize”, “Rigid-body fit”, “Rotate/translate”, “Rotamer” and “Torsion editing” as implanted in COOT 0.9.4.1.

4. What algorithm is implemented in the real space refinement in the PHENIX software?

Response: Thanks for the question. As described in both literature (PMIDs: 29872004 and 30198894) and online documentation (https://phenix-online.org/documentation/reference/real_space_refine.html; <https://phenix-online.org/documentation/overviews/cryo-em-real-space-refinement.html>), there are multiple algorithms involved in the real space refinement (*phenix.real_space_refine*) in the PHENIX. Basically, the real space refinement tool aims at obtaining a model that fits the map as good as possible while possessing a meaningful geometry (no validation outliers, such as Ramachandran plot or rotamer outliers). A target function guides the refinement by linking the model parameters to the experimental data and by scoring the model-versus-data fit. For cryo-EM data, refinement of the model is done in real space and the target function is formulated in terms of a three-dimensional map. Because there are generally too many model parameters, refinement requires additional restraints that modify the target function by creating relationships between independent parameters.

Specifically, the real space refinement tool begins by reading a model file, map data and other parameters such as resolution information and/or additional restraint for ligands. Then, the tool proceeds to calculations that constitute a set of tasks repeated multiple times (macro-cycles). Tasks to be performed during the refinement that combine several algorithms including gradient-driven minimization of the entire model, simulated annealing, morphing, rigid-body refinement and local grid search. With the help of PHENIX comprehensive validation program making extensive use of the MolProbity validation algorithms, we could perform further optimization based on a detailed report on model quality and model-to-data fit (PMIDs: 29872004 and 31588918).

5. For the structures C4 and DC591053 bound to RXFP4-G_i, the initial structure of the complex INSL5-RXFP4-G_i was used (line 523). May the authors clarify the modeling process of compound 4 and DC591053, and how the atom positions were assigned for the synthetic agonists?

Response: Thanks for the comment. In the modeling process of compound 4 and DC591053-bound RXFP4-G_i complexes, we first prepared the ligand structures (in PDB format) and restraints (in CIF format) for these two synthetic agonists by eLBOW in PHENIX v1.18, then the ligand coordinates (in PDB format) were fitted to the cryo-EM density by LigandFit GUI (PMID: 16855309) in PHENIX v1.18. Meanwhile, the coordinates of the RXFP4-G_i complex obtained from the structural model of INSL5-RXFP4-G_i by removing the coordinates of INSL5 were docked to the EM density map using UCSF Chimera v1.13.1. The combinations of the fitted ligands (compound 4 or DC591053) and the RXFP4-G_i complex were prepared as starting points for further structure refinement in PHENIX v1.18 and manually adjusted and rebuilt in COOT 0.9.4.1.

In the molecular dynamics simulation section authors describe a setup for the INSL5-RXFP4 complex in a POPC lipid bilayer.

1. May the authors provide more context on the purpose of the MD simulation study? Was it designed to test the consistency of the interactions found in the resolved structure against those found in a MD all-atom Force Field calculation?

Response: We thank the reviewer for this important concern. Molecular dynamics (MD) simulation can provide a unique insight into the dynamic properties of GPCRs in a way that is complementary to many experimental approaches (PMID: 29188561). Herein, exactly as the reviewer pointed out, the MD simulation of INSL5-RXFP4 was performed to examine

both the overall stability of the INSL5–RXFP4 complex as well as their detailed residue-level interactions, thereby highlighting the key interactions that maintain peptide recognition (Supplementary Figure 8). Besides, to explore the structural stability of peptide, we performed MD simulations of INSL5 (including both A and B chains) and its B chain only (Supplementary Figure 9). The MD simulations were repeated independently three times with similar results.

2. From figures 8 and 9 in Suppl-info, there is a shift of the INSL5 in the receptor binding pocket. As the receptor binding pocket seems well preserved (Suppl Figure 8c) What interactions produce the C-terminus motion of W24? Did authors consider water mediated interactions or hydration level in the interhelical region? Reports of internal water molecules in crystal structures in GPCRs are known in literature.

Response: We appreciate the reviewer’s valuable comment. The motions of INSL5 B chain C terminus toward TM5 have been observed in all MD simulation trajectories, where its carboxyl end (COO- in W24^B) forms two hydrogen bonds with Q205 and one salt bridge with R208 as shown in Supplementary Figure 8d. Exactly as the reviewer pointed out, after analyzing the MD simulation trajectories, the internal waters were found to fill the orthosteric pocket with formation of extensive contacts with the surrounding polar residues from both RXFP4 (such as E100, R208, S254, N262, H263, T294, T295, H299, N301 and N305) and INSL5 (including R23^B and W24^B). Such hydration certainly advances the movements of the INSL5 B chain C terminus towards the TM5 because of the beneficial solvation energy and entropy.

The Supplementary Figure 8 (see below) has been significantly expanded by adding two new panels (g and h) to highlight the internal water molecules in the orthosteric pocket. The following statement and relevant literature have been added to the manuscript: “Consistently, molecular dynamics (MD) simulations found that the C-terminal α -helix of the B chain could stably maintain its insertion into the orthosteric pocket through its tip residues, evidenced by the interface area and representative minimum distances (R13^B–D104^{2.67}, R23^B–E100^{2.63} and W24^B–Q205^{5.39}/R208^{5.42}) (Supplementary Fig. 8a-f). Notably, the internal water molecules were found to fill the orthosteric pocket with the formation of multiple contacts with surrounding polar residues in both RXFP4 and the C terminus of INSL5 B chain during MD simulations (Supplementary Fig. 8g, h) as seen in other GPCRs³⁶⁻³⁸.”

Supplementary Figure 8. MD simulations of INSL5–bound active RXFP4. a, Comparison of the INSL5 conformation between the final simulation snapshot at 1,000 ns and the cryo-EM structure of INSL5–RXFP4–G_i complex. The key

residues in the peptide-receptor interface are shown in sticks. **b-d**, Close-up views of the interactions between the C terminal α -helix of INSL5 B chain and receptor residues and their minimum distances during MD simulations. **e**, RMSD of C α positions of the RXFP4 and INSL5, where all snapshots were superimposed on the cryo-EM structure of RXFP4 and INSL5 using the C α atoms, respectively. **f**, The interface area between RXFP4 and INSL5 (blue) or the two C terminus residues R23^B and W24^B (red), calculated by freeSASA 2.0. The thick and thin traces represent moving averages and original, unsmoothed values obtained from one single MD simulation trajectory, respectively. **g**, The distribution of water molecules in the orthosteric pocket that overlap with the position of indole group of W24^B in the cryo-EM structure model (green). A close-up view of the internal water molecules distributed around the C terminus of the INSL5 B chain was shown on the right. **h**, Radial distribution functions (RDFs, black) and cumulative number (blue) of water molecules around the C terminus of the INSL5 B chain. RDFs were computed from the oxygen atoms of water molecules to the non-hydrogen (heavy) atoms of W24^B for the last 500 ns MD simulation trajectory. The MD simulations were repeated independently three times with similar results.

3. Authors may include more details of the simulation protocol for reproducibility: I. system setup, II. equilibration and production protocols III analysis for data collection. Calculation of the binding free energy could be relevant to demonstrate selectivity of DC591053 in the RXFP4-Gi complex. Indeed, in line 183 of the main text authors mention that “C-terminal α -helix of the B-chain could stably insert into the orthosteric pocket through its tip residues”. This statement is misleading as the starting configuration included already the INSL5 in the binding pocket. A description of the insertion mechanism would require the calculation of the transition from unbound to bound states using, for example, umbrella sampling.

Response: We totally agreed with the above comments. To increase the reproducibility, the detail MD simulations steps were added to the manuscript as Supplementary Table 6, while the starting configuration of the MD simulations generated by the CHARMM-GUI webserver and all necessary input files have been uploaded to the submission system as a supporting file named “MDinputs-RXFP4.zip”.

As the reviewer pointed out, the starting configuration of MD simulation is built on the cryo-EM structure, where INSL5 was already inserted into the binding pocket. The MD simulations verified that the binding between INSL5 and RXFP4 is stable with the formation of multiple polar contacts. To avoid potential misunderstanding, the corresponding statement has been revised as “Consistently, molecular dynamics (MD) simulations found that the C-terminal α -helix of the B chain could stably **maintain its** insertion into the orthosteric pocket through its tip residues, evidenced by the interface area and representative minimum distances (R13^B–D104^{2.67}, R23^B–E100^{2.63} and W24^B–Q205^{5.39}/R208^{5.42}) (Supplementary Fig. 8a-f).”

Supplementary Table 6. Details of restraints applied during MD simulations.

Stage	Time step	Simulation time	Restrain
Heating	1 fs	1 ns	Position harmonic restrain (40 kJ·mol ⁻¹ ·Å ⁻²) for the backbone non-hydrogen atoms of protein and peptide; Position restrain (20 kJ·mol ⁻¹ ·Å ⁻²) for the sidechain non-hydrogen atoms of protein and peptide; Planar harmonic restraint (10 kJ·mol ⁻¹ ·Å ⁻²) for the phosphorus atom of POPC along the Z-axis; Dihedral restraint (1000 kJ·mol ⁻¹ ·rad ⁻²) for two dihedrals (C28-C29-C210-C211 and C1-C3-C2-O21).
Step6.1	1 fs	5 ns	Position harmonic restrain (40 kJ·mol ⁻¹ ·Å ⁻²) for the backbone non-hydrogen atoms of protein and peptide; Position restrain (20 kJ·mol ⁻¹ ·Å ⁻²) for the sidechain non-hydrogen atoms of protein and peptide;

			Planar harmonic restraint ($10 \text{ kJ}\cdot\text{mol}^{-1}\cdot\text{\AA}^{-2}$) for the phosphorus atom of POPC along the Z-axis; Dihedral restraint ($1000 \text{ kJ}\cdot\text{mol}^{-1}\cdot\text{rad}^{-2}$) for two dihedrals (C28-C29-C210-C211 and C1-C3-C2-O21).
Step6.2	1 fs	5 ns	Position harmonic restrain ($20 \text{ kJ}\cdot\text{mol}^{-1}\cdot\text{\AA}^{-2}$) for the backbone non-hydrogen atoms of protein and peptide; Position restrain ($10 \text{ kJ}\cdot\text{mol}^{-1}\cdot\text{\AA}^{-2}$) for the sidechain non-hydrogen atoms of protein and peptide; Planar harmonic restraint ($4 \text{ kJ}\cdot\text{mol}^{-1}\cdot\text{\AA}^{-2}$) for the phosphorus atom of POPC along the Z-axis; Dihedral restraint ($400 \text{ kJ}\cdot\text{mol}^{-1}\cdot\text{rad}^{-2}$) for two dihedrals (C28-C29-C210-C211 and C1-C3-C2-O21).
Step6.3	2 fs	10 ns	Position harmonic restrain ($10 \text{ kJ}\cdot\text{mol}^{-1}\cdot\text{\AA}^{-2}$) for the backbone non-hydrogen atoms of protein and peptide; Position restrain ($5 \text{ kJ}\cdot\text{mol}^{-1}\cdot\text{\AA}^{-2}$) for the sidechain non-hydrogen atoms of protein and peptide; Planar harmonic restraint ($4 \text{ kJ}\cdot\text{mol}^{-1}\cdot\text{\AA}^{-2}$) for the phosphorus atom of POPC along the Z-axis; Dihedral restraint ($200 \text{ kJ}\cdot\text{mol}^{-1}\cdot\text{rad}^{-2}$) for two dihedrals (C28-C29-C210-C211 and C1-C3-C2-O21).
Step6.4	2 fs	10 ns	Position harmonic restrain ($5 \text{ kJ}\cdot\text{mol}^{-1}\cdot\text{\AA}^{-2}$) for the backbone non-hydrogen atoms of protein and peptide; Position restrain ($2 \text{ kJ}\cdot\text{mol}^{-1}\cdot\text{\AA}^{-2}$) for the sidechain non-hydrogen atoms of protein and peptide; Planar harmonic restraint ($2 \text{ kJ}\cdot\text{mol}^{-1}\cdot\text{\AA}^{-2}$) for the phosphorus atom of POPC along the Z-axis; Dihedral restraint ($200 \text{ kJ}\cdot\text{mol}^{-1}\cdot\text{rad}^{-2}$) for two dihedrals (C28-C29-C210-C211 and C1-C3-C2-O21).
Step6.5	2 fs	10 ns	Position harmonic restrain ($2 \text{ kJ}\cdot\text{mol}^{-1}\cdot\text{\AA}^{-2}$) for the backbone non-hydrogen atoms of protein and peptide; Position restrain ($0.5 \text{ kJ}\cdot\text{mol}^{-1}\cdot\text{\AA}^{-2}$) for the sidechain non-hydrogen atoms of protein and peptide; Planar harmonic restraint ($0.4 \text{ kJ}\cdot\text{mol}^{-1}\cdot\text{\AA}^{-2}$) for the phosphorus atom of POPC along the Z-axis; Dihedral restraint ($100 \text{ kJ}\cdot\text{mol}^{-1}\cdot\text{rad}^{-2}$) for two dihedrals (C28-C29-C210-C211 and C1-C3-C2-O21).
Step6.6	2 fs	10 ns	Position harmonic restrain ($0.5 \text{ kJ}\cdot\text{mol}^{-1}\cdot\text{\AA}^{-2}$) for the backbone non-hydrogen atoms of protein and peptide;
Step7	2 fs	1000 ns	Restrain-free

In section Expression and purification of the RXFP4-Gi complex describes the procedure to produce the recombinant receptors in insect cells. May the authors briefly describe the procedure to produce the receptor mutants?

Response: We thank the reviewer for this suggestion. In the functional assays, human RXFP4 (NCBI Reference Sequence: NM_181885.3) and RXFP3 (NCBI Reference Sequence: NM_016568.3) were cloned into pCMV6 constructs (OriGene Technologies). The mutant receptors were modified by site-directed mutagenesis in the setting of the WT constructs, with the primers designed by QuikChange Primer Design [(QuikChange Primer Design (<http://agilent.com.cn>))] and carried

out using Phanta Max Master (Vazyme). Sequences of all primers used in this study were provided in Supplementary Table 5, and all the constructs were confirmed by DNA sequencing. The constructs were transfected into H293T cells using Lipofectamine 3000 transfection reagent (Invitrogen) and the receptor mutants were ready for experiment following 24 h culturing. This information has been added to the Construct section of methods: “The full-length human RXFP4 (NCBI Reference Sequence: NM_181885.3) was cloned into a modified pFastBac vector (Invitrogen) with HA signal peptide to enhance receptor expression, followed by a 10× histidine tag and BRIL insertion at the N terminus. LgBiT subunit (Promega) was fused at the C terminus of RXFP4 connected by a 15-amino acid polypeptide linker. A dominant-negative human $G\alpha_{i2}$ (DNGi2) were generated by introducing S47N, G203A, E245A and A326S substitutions in the $G\alpha$ subunit as previously described⁶⁹. The human $G\beta 1$ with a C-terminal 15-amino acid polypeptide linker was followed by a HiBiT (peptide 86, Promega), and the scFv16 was modified with an N-terminal GP67 signaling peptide and a C-terminal 8× histidine tag. The engineered human $G\alpha_{i2}$, $G\beta 1$, bovine $G\gamma 2$ and scFv16 were cloned into the pFastBac vector (Invitrogen), respectively. For cAMP accumulation assay, human RXFP4 and RXFP3 (NCBI Reference Sequence: NM_016568.3) were cloned into pCMV6 constructs (OriGene Technologies). The mutant receptors were modified by site-directed mutagenesis in the setting of the WT constructs, with the primers designed by QuikChange Primer Design [QuikChange Primer Design (<http://agilent.com.cn>)] and carried out using Phanta Max Master (Vazyme). N-terminal Flag tag was added to both WT and mutant receptors for surface expression measurement. Sequences of all primers used in this study were provided in **Supplementary Table 5**, and all the constructs were confirmed by DNA sequencing.”

Supplementary Table 5. Primers used in this study, related to Figures 2, 3, Supplementary Figures 3, 10, and Supplementary Tables 3 and 4.

Oligonucleotide name	Oligonucleotide sequence (5'-3')	Cloning method	Product
Insert-fragment-forward	AGATCTGCGCCGCGATCGCCAAAATGAAGAC GATCATCGCC	Homologous recombination	pCMV6-HA-H10-BRIL-RXFP4(1-374)-LgBiT
Insert-fragment-reverse	CTATGACCGCGCCGGCCGTTTAGCTGTTGATG GTTACTCGGAA		
Linear-pCMV6-forward	ACGGCCGGCCGCGGTCAT		
Linear-pCMV6-reverse	GGCGATCGCGGCAGAT		
Add-RXFP4-Flag-forward	CGATCGCCATGGACTACAAAGACGATGACGAC AAGCCCACTCAATACT	Site-directed mutagenesis	pCMV6-Flag-RXFP4
Add-RXFP4-Flag-reverse	GAGTGTGGGCTTGTCGTCATCGTCTTTGTAGTC CATGGCGATCGCGGCGC		
Add-RXFP3-Flag-forward	CTTGCCATGGACTACAAAGACGATGACGACAA GCAGATGGCCGATGCAGCCAC		pCMV6-Flag-RXFP3
Add-RXFP3-Flag-reverse	CTTGTCGTCATCGTCTTTGTAGTCCATGGCAAG CTTGGCGGCAGATCTC		
W97A-forward	GCACTACTCTCCCCTTTGCGGCAGCCGAG	Site-directed mutagenesis	pCMV6-RXFP4(1-374)-W97A
W97A-reverse	CTCGGCTGCCGCAAAGGGGAGAGTGAGTGC		
E100A-forward	TTTTGGGCAGCCGCGTCGGCACTGGAC		pCMV6-RXFP4(1-374)-E100A
E100A-reverse	GTCCAGTGCCGACGCGGCTGCCAAAA		
D104A-forward	GAGTCGGCACTGGCCTTTCCTGGCCC		pCMV6-RXFP4(1-374)-D104A
D104A-reverse	GGCCAGTGAAAGGCCAGTGCCGACTC		

F105A-forward	GAGTCGGCACTGGACGCTCACTGGCCCTTCGG	pCMV6-RXFP4(1-374)-F105A
F105A-reverse	CCGAAGGGCCAGTGAGCGTCCAGTGCCGACTC	
T121A-forward	TGGTTCTGACGGCCGCTGTCTCAACGT	pCMV6-RXFP4(1-374)-T121A
T121A-reverse	GACGTTGAGGACAGCGGCCGTCAGAACCA	
R194A-forward	GCCTTTGCCTGCTGGCTTTCCCCAGCAGGT	pCMV6-RXFP4(1-374)-R194A
R194A-reverse	ACCTGCTGGGGAAAGCCAGCAGGCAAAGGC	
Q205A-forward	GCTGGGGGCTACGCGCTGCAGAGGGTG	pCMV6-RXFP4(1-374)-Q205A
Q205A-reverse	CACCCTCTGCAGCGGTAGGCCCCAGC	
R208A-forward	CCTACCAGCTGCAGGCGGTGGTGCTGGCTT	pCMV6-RXFP4(1-374)-R208A
R208A-reverse	AAGCCAGCACCAACCGCCTGCAGCTGGTAGG	
K273A-forward	TGGGGTGTCTGGTGGCGTTTGACCTGGTGCC	pCMV6-RXFP4(1-374)-K273A
K273A-reverse	GGCACCAGGTCAAACGCCACCAGGACACCCCA	
W279A-forward	GAAGTTTGACCTGGTGCCCGCAACAGTACTTTCTATACTA	pCMV6-RXFP4(1-374)-W279A
W279A-reverse	TAGTATAGAAAGTACTGTTTCGCGGGCACCAGGTCAAACCTC	
Y284A-forward	GCCCTGGAACAGTACTTTTCGCTACTATCCAGACGTATGTC	pCMV6-RXFP4(1-374)-Y284A
Y284A-reverse	GACATACGTCTGGATAGTAGCGAAAGTACTGTTCCAGGGC	
T295A-forward	TGTCTTCCCTGTCACTGCTTGCTTGGCACACAG	pCMV6-RXFP4(1-374)-T295A
T295A-reverse	CTGTGTGCCAAGCAAGCAGTGACAGGGAAGACA	
H299A-forward	GTCACTACTTGCTTGGCAGCCAGCAATAGCTGCTCAA	pCMV6-RXFP4(1-374)-H299A
H299A-reverse	TTGAGGCAGCTATTGCTGGCTGCCAAGCAAGTAGTGAC	
Q205H-forward	GGGGCCTACCATCTGCAGAGGGTG	pCMV6-RXFP4(1-374)-Q205H
Q205H-reverse	CACCCTCTGCAGATGGTAGGCCCC	
R208K-forward	GGCCTACCAGCTGCAGAAGGTGGTGCT	pCMV6-RXFP4(1-374)-R208K
R208K-reverse	AGCACCACCTTCTGCAGCTGGTAGGCC	
T295V-forward	GCCCTCTGCAAGATGGTTTCGACGGCCACTAGCTCAACGTCTATGC	pCMV6-RXFP4(1-374)-T295V
T295V-reverse	GCATAGACGTTGAGGCTAGTGCCGTCGAAACCATCTTGCAGAGGGC	
L118S+V122S-forward	GCCCTCTGCAAGATGGTTTCGACGGCCACTAGCTCAACGTCTATGC	

L118S+V122S-reverse	GCATAGACGTTGAGGCTAGTGGCCGTCGAAAC CATCTTGCAGAGGGC		pCMV6- RXFP4(1-374)- L118S+V122S
H268Q-forward	TGGCTGGGCCTCTACCAGTCGCAGAAG	Site-directed mutagenesis	pCMV6- RXFP3(1-469)- H268Q
H268Q-reverse	CTTCTGCGACTGGTAGAGGCCAGCCA		pCMV6- RXFP3(1-469)- K271R
K271R-forward	GGCCTCTACCACTCGCAGAGGGTGCTGCTG		pCMV6- RXFP3(1-469)- V375T
K271R-reverse	CAGCAGCACCTCTGCGAGTGGTAGAGGCC		pCMV6- RXFP3(1-469)- S159L+S163V
V375T-forward	GCGTTCCTGTGAGCACGTGCCTAGCGCACTC		
V375T-reverse	GAGTGCCTAGGCACGTGCTCACAGGGAACGC		
S159L+S163V-forward	GCCATGTGTAAGATCGTGTTAATGGTGACGGTC ATGAACATGTACGCCAGC		
S159L+S163V-reverse	GCTGGCGTACATGTTTCATGACCGTCACCATTAA CACGATCTTACACATGGC		

For structure validation it is recommended to report the Rama-Z score (Structure 28, 1249–1258.e1–e2, November 3, 2020), which provides a criterion for improbable backbone geometry $|Z|>3$, $2<|Z|<3$ possible geometry, and $|Z|<2$ for normal geometry. In Suppl-Table 1, the favorable, allowed and outliers was reported as a percentage, which is not a definitive criterion for a good shape of the Ramachandran angles distribution.

Response: We appreciate this valuable suggestion. The Rama-Z scores of the whole complexes [0.71 for INSL5–RXFP4–G_i, 0.16 for Compound 4–RXFP4–G_i and 0.78 for DC591053–RXFP4–G_i] were included in the revised Supplementary Table 1 and the reference (Structure 28, 1249–1258.e1–e2, November 3, 2020) has been cited in the manuscript.

On lines 51-52 authors describe differences in the N-terminal tail, between RXFP1-2 and RXFP3-4, and mention 43% of sequence identity. Could authors clarify whether the sequence identity refers to the TM domains, the N-terminal, or was it for the overall structure?

Response: We thank the reviewer for the valuable suggestion and the corresponding sentence has been revised as: “RXFP3 and RXFP4 have distinct binding properties with a-relatively short N-terminal tails rather than LRR. They possess 43% sequence identity in the overall structure and inhibit cAMP production via pertussis toxin-sensitive G_{α_{i/o}} proteins.”

Reviewers' Comments:

Reviewer #1:

Remarks to the Author:

The authors generally addressed my concerns. Some minor points should be addressed before publication.

1. Data for the major point 1 can be added into the paper.
2. Orthostetic in line 32 should be orthosteric.
3. The sentence "Because of high cross-reactivity, it can not be used therapeutically" should be deleted since no evidence supports this argument.

Reviewer #2:

Remarks to the Author:

MS Title: A unique peptide recognition mechanism by the human relaxin family peptide receptor 4 (RXFP4)

Dear Authors:

The revised version of the manuscript, including supplementary information and answers to reviewers, largely clarify all of my comments and observations. This work represents a step forward in understanding mechanisms of receptor activation in the context of the GPCR superfamily. The only two points that may help to improve the presentation of results in the MD section are:

1. Supplementary figure 8, panel h, shows a radial distribution function that is not well normalized. Since $g(r)$ represents a conditional probability, for large values of r (after third hydration shell in bulk water) it should reach a value of one. The $g(r)$ plot, as here presented, does not provide any meaningful information. Solvation of W24 could be driven by the carboxyl terminus, while the indole ring could promote hydrophobic effect. Thus, positional and orientational correlations of water molecules solvating W24 in the binding pocket of RXFP4 may require more extensive sampling. As an alternative, if necessary, plots for the number of water molecules at 2-3 Å from the W24 (for example, shown as time series) can be reported, i.e., picking only water molecules forming the first "solvation" shell. From such analysis it is possible to detect persistent or conserved water-W24 interactions.
2. Supplementary Table 6 shows the steps for pre-equilibration of the system. Is there any plot that authors may show to determine the system stability? Often the box dimensions over time helps to determine stability of the simulation box, the XY-area and Z-height; or rmsd of the Ca atoms of the protein reaching a stable average over time.

POINT-BY-POINT RESPONSES TO THE REVIEWERS' COMMENTS

Reviewer #1 (Remarks to the Author):

The authors generally addressed my concerns. Some minor points should be addressed before publication.

1. Data for the major point 1 can be added into the paper.

Response: This point is well taken. These experimental data have been added to the manuscript as Supplementary Figure 10e-g (see below). To reflect this, the following statement has been added to the manuscript: “To further explore subtype selectivity, we performed amino acid switch studies in the equivalent positions between RXFP4 and RXFP3 around the ligand-binding pocket. Double mutant L118^{3.29}S+V122^{3.33}S in RXFP4 selectively affected the potency of DC591053 by 20.9-fold without notable influence on that of compound 4. As a comparison, S159^{3.29}L+S163^{3.33}V in RXFP3 reduced the potency of compound 4 by 26.9-fold. Similar phenomena were also observed in Q205^{5.39}H and R208^{5.42}K in RXFP4 (displayed more profound reduction for DC591053 than compound 4), while H268^{5.39}Q and K271^{5.42}R in RXFP3 exhibited dose-response features for compound 4 similar to the WT (Supplementary Fig. 10b-d, Supplementary Table 4). Notably, mutations at S159^{3.29}, S163^{3.33} and V375^{7.39} in RXFP3 and L118^{3.29}, V122^{3.33} and T295^{7.39} in RXFP4 caused differentiated influences on the potencies of INSL5 and relaxin-3 (Supplementary Fig. 10e-g, Supplementary Table 5). The results indicate that these sites may play important roles in subtype selectivity.”

Supplementary Figure 10. Peptidomimetic agonism and key residues on receptor subtype selectivity of compound 4 and DC591053. a, RXFP4 residues are categorized according to their interactions with the three ligands. b-d, Effects of amino acid switch in equivalent positions between

RXFP4 and RXFP3 around the ligand-binding pocket on compound 4 (**b**) and DC591053 (**c**) induced cAMP accumulation in RXFP4 as well as on compound 4 induced cAMP accumulation in RXFP3 (**d**). **e, f**, Effects of INSL5 (**e**) and relaxin-3 (**f**) on cAMP accumulation in wild-type (WT) and mutant RXFP4s. **g**, Effects of relaxin-3 on cAMP accumulation in WT and mutant RXFP3. INSL5 was totally inactive in RXFP4 single mutants L118^{3.29}S and L118^{3.29}A as well as double mutants L118^{3.29}S+V122^{3.33}S and L118^{3.29}A+V122^{3.33}A, where relaxin-3 retained partial activity although the curves shifted to the right (by 3.2-fold, 5.4-fold, 21.8-fold and 9.7-fold, respectively). For comparison, relaxin-3 activated S159^{3.29}A, S159^{3.29}L, S159^{3.29}L+S163^{3.33}V and S159^{3.29}A+S163^{3.33}A in RXFP3 albeit with reduced potencies. T295^{7.39}V did not destroy the response of RXFP4 to INSL5 and relaxin-3, but V375^{7.39}T in RXFP3 impaired both the potency (by 5.1-fold) and E_{max} (66.5% of the WT) of relaxin-3 (INSL5 was inactive in WT and all the five RXFP3 mutants). Therefore, S159^{3.29}, S163^{3.33} and V375^{7.39} in RXFP3 and L118^{3.29}, V122^{3.33} and T295^{7.39} in RXFP4 are likely involved in RXFP3 vs. RXFP4 subtype selectivity, consistent with the observations in RXFP3/RXFP4 chimeric receptor studies⁴. Data were shown as means ± S.E.M. of at least three independent experiments (~~*n* = 3-6~~). The numbers of independent experiments are shown in the parentheses. Supplementary Tables 4 and 5 provide detailed statistical evaluation such as *P* values and numbers of independent experiments (*n*). Source data are provided as a Source Data file.

2. Orthostetic in line 32 should be orthosteric.

Response: This point is well taken, thanks

3. The sentence "Because of high cross-reactivity, it can not be used therapeutically" should be deleted since no evidence supports this argument.

Response: This point is well taken, thanks

Reviewer #2 (Remarks to the Author):

MS Title: A unique peptide recognition mechanism by the human relaxin fam22ily peptide receptor 4 (RXFP4)

Dear Authors:

The revised version of the manuscript, including supplementary information and answers to reviewers, largely calrify all of my comments and observations. This work represent a step forward in understanding mechanisms of receptor activation in the context of the GPCR superfamily. The only two points that may help to improve the presentation of results in the MD section are:

1. Supplementary figure 8, panel h, shows a radial distribution function that is not well normalized. Since $g(r)$ represent a conditional probability, for large values of r (after third hydration shell in bulk water) it should reach a value of one. The $g(r)$ plot, as here presented, does not provide any meaningful information. Solvation of W24 could be driven by the carboxyl terminus, while the indole ring could promote hydrophobic effect. Thus, positional and orientational correlations of water molecules solvating W24 in the binding pocket of RXFP4 may require more extensive sampling. As an alternative, if necessary, plots for the number of water molecules at 2-3 Å from the W24 (for example, shown as time series) can

be reported, i.e., picking only water molecules forming the first "solvation" shell. From such analysis it is possible to detect persistent or conserved water-W24 interactions.

Response: We appreciate the reviewer's valuable comment. We plotted the time evolution of the number of water molecules within the cut-off distances (2.0 Å, 2.5 Å, 3.0 Å, 3.5 Å, and 4.0 Å) of W24^B during molecular dynamics (MD) simulation (Figure X1). These water molecules whose oxygen atoms located within the cut-off distance of at least one heavy atom in the W24^B were counted. The average numbers of water molecules within 2.0 Å, 2.5 Å, 3.0 Å, 3.5 Å, and 4.0 Å of W24^B during MD simulation are 0, 0.06, 4.56, 6.29, and 8.42, respectively. To reflect this, the Supplementary Figure 8 has been significantly expanded by adding these two panels (Figure X1) to reveal these conserved water-W24 interactions during MD simulation.

Figure X1. Distribution of water molecules in the orthosteric pocket of RXFP4 during MD simulation. **a**, The distribution of water molecules in the orthosteric pocket that overlap with the position of indole group of W24^B in the cryo-EM structure model (green). A close-up view of the internal water molecules distributed around the C terminus of the INSL5 B chain was shown on the right. **b**, Time evolution of the number of water molecules within the cut-off distance of W24^B during MD simulation. Only these water molecules whose oxygen atoms located within the cut-off distance of at least one heavy atom in the W24^B were counted. Five cut-off distances (2.0 Å, 2.5 Å, 3.0 Å, 3.5 Å, and 4.0 Å) were adopted. The MD simulations were repeated independently three times with similar results.

2. *Supplementary Table 6 shows the steps for pre-equilibration of the system. Is there any plot that authors may show to determine the system stability? Often the box dimensions over time helps to determine stability of the simulation box, the XY-area and Z-height; or rmsd of the Ca atoms of the protein reaching a stable average over time.*

Response: Thanks for the comment. Per the reviewer's suggestion, we calculated the Z-axis height and the XY plane area of the simulation box, the potential energy of the MD simulation system, the radius of gyration (RXFP4) and root mean squared deviation (RMSD) of C α positions of the RXFP4 during MD simulation as shown in Figure X2. These results demonstrate the high stability of MD simulation system: the MD simulation box was well maintained during MD simulation, and the simulated protein RXFP4 was well-equilibrated at the desired temperature and pressure. To reflect this, the Supplementary Figure 8 has been significantly expanded by adding these panels (Figure X2) to highlight the system stability during MD simulation.

Figure X2. Estimation of system stability during the MD simulations of INSL5-bound active RXFP4. **a**, Time evolution of the Z-axis height (top) and XY plane area (bottom) of the simulation box during MD simulation. **b**, Potential energy fluctuation during MD simulation. **c**, Radius of gyration (Rg) of RXFP4 during MD simulation. **d**, Root mean squared deviation (RMSD) of C α positions of the RXFP4 and INSL5, where all snapshots were superimposed on the cryo-EM structure of RXFP4 and INSL5 using the C α atoms, respectively.